# A low affinity *cis*-regulatory BMP response element restricts target gene activation to subsets of Drosophila neurons

Anthony JE Berndt[1†], Katerina M Othonos[2†], Tianshun Lian[2], Stephane Flibotte[3], Mo Miao[2], Shamsuddin A Bhuiyan[4], Raymond Y Cho[2], Justin S Fong[2], Seo Am Hur[2], Paul Pavlidis[4], Douglas W Allan[2]*

[1]Department of Food & Fuel for the 21st Century, University of California San Diego, San Diego, United States; [2]Department of Cellular and Physiological Sciences, University of British Columbia, Vancouver, Canada; [3]UBC/LSI Bioinformatics Facility, University of British Columbia, Vancouver, Canada; [4]Department of Psychiatry, University of British Columbia, Vancouver, Canada

**Abstract** Retrograde BMP signaling and canonical pMad/Medea-mediated transcription regulate diverse target genes across subsets of *Drosophila* efferent neurons, to differentiate neuropeptidergic neurons and promote motor neuron terminal maturation. How a common BMP signal regulates diverse target genes across many neuronal subsets remains largely unresolved, although available evidence implicates subset-specific transcription factor codes rather than differences in BMP signaling. Here we examine the *cis*-regulatory mechanisms restricting BMP-induced *FMRFa* neuropeptide expression to Tv4-neurons. We find that pMad/Medea bind at an atypical, low affinity motif in the *FMRFa* enhancer. Converting this motif to high affinity caused ectopic enhancer activity and eliminated Tv4-neuron expression. In silico searches identified additional motif instances functional in other efferent neurons, implicating broader functions for this motif in BMP-dependent enhancer activity. Thus, differential interpretation of a common BMP signal, conferred by low affinity pMad/Medea binding motifs, can contribute to the specification of BMP target genes in efferent neuron subsets.

*For correspondence:
doug.allan@ubc.ca

†These authors contributed equally to this work

Competing interests: The authors declare that no competing interests exist.

## Introduction

Diverse neuronal subtypes are generated by the gene regulatory activities of subtype-specific combinations of transcription factors (*Allan and Thor, 2015*; *da Silva and Wang, 2011*; *Hobert et al., 2010*). In postmitotic neurons, retrograde signaling continues to play prominent roles during neuronal maturation and through life in regulating terminal differentiation and synaptic function (*Allan et al., 2003*; *Angley et al., 2003*; *Veverytsa and Allan, 2011*; *Berke et al., 2013*; *Chou et al., 2013*; *Kelly et al., 2013*; *Majdazari et al., 2013*; *Xiao et al., 2013*; *DuVal et al., 2014*; *McCabe et al., 2003*; *Guha et al., 2004*; *Lopez-Coviella et al., 2005*; *Hodge et al., 2007*; *Pavelock et al., 2007*; *Miguel-Aliaga et al., 2008*; *Eade and Allan, 2009*; *Henríquez et al., 2011*). In *Drosophila*, retrograde BMP signaling occurs within most efferent neurons, regulating a diverse suite of target genes in a subtype-specific manner; for example, the neuropeptide genes unique to each neurosecretory subtype, and the partially overlapping battery of genes that support the growth and strengthening of motor neuron neuromuscular junction synapses (*Veverytsa and Allan, 2011*; *Miguel-Aliaga et al., 2008*; *Allan et al., 2003*; *Kim and Marqués, 2010*; *Vuilleumier et al., 2019*; *Ball et al., 2010*; *Marqués et al., 2003*). An unresolved question regards how BMP target genes are

differentially specified within each of these BMP-activated neuron subtypes. Prevailing models suggest that subtype-specific transcription factors play the primary role in target gene selection, and BMP-responsive DNA motifs within those target gene enhancers encode generic responsiveness to BMP signaling in any neuron (*Veverytsa and Allan, 2011*; *Miguel-Aliaga et al., 2008*; *Allan et al., 2005*). Here we describe our identification of a low affinity BMP-responsive DNA motif that instructively contributes to the differential specification of BMP target genes within efferent neuron subtypes.

In *Drosophila* efferent neurons, retrograde BMP-signaling is triggered by Glass bottom boat (Gbb) ligand engagement at a presynaptic BMP-Receptor (BMP-R) complex of Wishful thinking (Wit), Thickveins (Tkv), and Saxophone (Sax). This leads to phosphorylation of Mad to pMad (vertebrate SMAD1/5/9) (*Allan et al., 2003*; *McCabe et al., 2003*; *Aberle et al., 2002*; *Rawson et al., 2003*), which then complexes with Medea (vertebrate SMAD4) (*Gao and Laughon, 2007*; *Gao et al., 2005*). This pMad/Medea complex translocates to the nucleus and regulates transcription by binding DNA in a sequence-specific manner at BMP-response elements (BMP-REs) (*Kim et al., 1997*; *Xu et al., 1998*; *Shi and Massagué, 2003*). In the nucleus, Schnurri (Shn) can act as a co-repressor with the pMad/Medea complex at silencer BMP-REs (BMP-SE) to repress target genes (*Gao et al., 2005*; *Pyrowolakis et al., 2004*; *Charbonnier et al., 2015*). Additionally, Brinker (Brk) is a DNA binding default repressor of genes in the absence of BMP-signaling (*Barolo and Posakony, 2002*; *Affolter and Basler, 2007*; *Hamaratoglu et al., 2014*).

Studies of BMP-dependent morphogenesis in the *Drosophila* embryo and wing imaginal disc have established core principles and sequence preferences for BMP-REs and transcriptional regulators of the BMP pathway. BMP-dependent gene activation is mediated directly by pMad/Medea acting at a bipartite 15 bp BMP-AE; $GGCGCCA(N_4)GNCV$ (*Weiss et al., 2010*; *Chayengia et al., 2019*). BMP-dependent gene silencing is mediated directly by pMad, Medea, and Shn acting in a complex at a bipartite 15 bp BMP-SE; $GRCGNC(N_5)GNCT$ (*Gao et al., 2005*; *Pyrowolakis et al., 2004*; *Charbonnier et al., 2015*). In these complexes, two pMad proteins bind to the 6 bp GC-rich sequence, and a single Medea protein binds to the 4 bp GNCV/GNCT sequence (*Gao et al., 2005*). An additional widely-utilized mechanism for BMP target gene activation in *Drosophila* involves *brk*-dependent de-repression (*Affolter and Basler, 2007*). Here BMP signaling results in pMad, Medea and Shn binding to multiple BMP-SE motifs in the *brk* locus (*Pyrowolakis et al., 2004*; *Marty et al., 2000*; *Torres-Vazquez et al., 2000*; *Müller et al., 2003*). Subsequent reduction in Brk levels relieves its default repression of many genes, which thereby become de-repressed in a BMP-dependent manner (*Pyrowolakis et al., 2004*; *Jaźwińska et al., 1999*; *Campbell and Tomlinson, 1999*; *Winter and Campbell, 2004*; *Minami et al., 1999*; *Moser and Campbell, 2005*).

We recently demonstrated that the BMP-AE motif serves as a widely-deployed BMP-dependent activator of gene expression in larval *Drosophila* efferent neurons (*Vuilleumier et al., 2019*). In that study, BMP-AE motifs were shown to confer BMP-dependence on enhancers that were active in diverse patterns across the many subtypes of efferent neurons. This diversity did not appear to be encoded by sequence differences within BMP-AEs. For example, enhancers that included similar 15 bp BMP-AE sequences differed greatly in expression pattern (such as Van 50 and Van 27 reporters), and swapping BMP-AE sequences between enhancers failed to change expression pattern (*Vuilleumier et al., 2019*). This notion is supported by recent work in *Drosophila* follicular epithelium and wing imaginal disc showing that the *cis*-regulatory environment around BMP-AE motifs, bound by other transcription factors, determines the enhancer's spatiotemporal activity, while the BMP-AE itself acts as a 'monotonic', or generic, interpreter of BMP activity (*Chayengia et al., 2019*).

The integration of BMP-activated pMad/Medea with subtype-specific transcription factors in *Drosophila* neurons has only been examined in detail in Tv4-neurons (*Chayengia et al., 2019*). Here a Tv4-neuron-specific transcription factor code and pMad/Medea were found to bind within a Tv4neuron-specific enhancer of the *FMRFa* gene (*Allan et al., 2003*; *Allan et al., 2005*; *Miguel-Aliaga et al., 2004*; *Berndt et al., 2015*; *Benveniste et al., 1998*), at a minimal 25 bp Homeodomain-Responsive Element, and a minimal 39 bp BMP-Responsive Element, respectively (*Berndt et al., 2015*). Misexpression of combinations of these transcription factors in other efferent neurons (with activated retrograde BMP signaling) led to widespread ectopic *FMRFa* gene activation (*Allan et al., 2003*; *Eade and Allan, 2009*; *Allan et al., 2005*; *Miguel-Aliaga et al., 2004*; *Berndt et al., 2015*; *Benveniste et al., 1998*; *Baumgardt et al., 2009*). By contrast, neither ectopic activation nor reduction in BMP signaling was found to alter Tv4-specific *FMRFa* expression. A

parsimonious model emerging from these studies would suggest that activator BMP-RE motifs add a generic BMP-responsive input for an enhancer whose spatiotemporal activity is otherwise determined by subtype-specific transcription factors.

Here we provide evidence for a second model, in which low affinity BMP-RE motifs instructively contribute to subtype-specific gene expression within subsets of BMP-activated efferent neurons. By functional and biochemical dissection of the 39 bp BMP-responsive *cis*-element of the *FMRFa* enhancer, we identified a non-canonical minimal pMad/Medea binding motif [*GGCGCC(N₅)GTAT*], which has low affinity (LA) compared to BMP-AE and BMP-SE motifs, that we herein term BMP-LA. Our genetic analysis showed that pMad and Medea act through this motif to directly activate *FMRFa* expression, without any contribution from *brinker* or *schnurri*. Notably, by converting this motif into a high affinity BMP-AE motif, we observed ectopic expression of BMP-dependent reporter activity into other neuronal populations, and a loss of expression in the Tv4-neuron itself. Finally, *in silico* searches and subsequent enhancer activity analysis revealed other functional instances of this motif in the genome. Overall, our results identify a novel low affinity BMP-RE motif type (BMP-LA), and lead to our proposal that the relative strength of BMP-responsive motifs is utilized to confer subtype-specific expression of BMP target genes in neurons.

## Results

A 445 bp enhancer region mediates Tv4-neuron-specific expression of the *FMRFa* gene (*Benveniste et al., 1998*; *Benveniste and Taghert, 1999*). We recently sub-mapped this enhancer and defined two short *cis*-elements that are together necessary for enhancer activity, but are not individually sufficient (*Berndt et al., 2015*). Toward defining the information encoded by each *cis*-element, we found that concatemers for both *cis*-elements independently generate highly specific Tv4-neuron activity in late embryos. Thus, both contain sufficient information for Tv4-specific activity. This allowed us to explore their transcription factor inputs, leading to their definition as a homeodomain-responsive *cis*-element (HD-RE) and a BMP-responsive *cis*-element (BMP-RCE). The HD-RE recruits and is activated by the LIM-homeodomain transcription factor, Apterous, but does not require BMP-signaling for activation. The BMP-RCE requires BMP-signaling for its activity and recruits pMad sequence-specifically at a *GGCGCC* site (*Berndt et al., 2015*; *Figure 1A*).

### The 15 bp *FMRFa* BMP-responsive *cis*-element contains a novel pMad/Medea-binding motif

Here we explored how the 39 bp BMP-RCE encodes activity that is BMP-dependent and Tv4-neuron-specific. We previously defined the necessity of a palindromic *GGCGGC* pMad-binding sequence, but not its associated Medea-binding sequence (*Berndt et al., 2015*). Here we find two *GNCV*-types sequences 5′ of the *GGCGCC* sequence; spaced 3 and 11 nucleotides away (green highlight in *Figure 1B*). Although not of canonical 5nt length, BMP-AE motifs with altered linker lengths have been shown to retain activity (*Weiss et al., 2010*; *Chayengia et al., 2019*; *Esteves et al., 2014*). We substitution mutagenized the 15 bp region containing these sequences, within the context of a full-length 445 bp Tv4-enhancer (*Figure 1—figure supplement 1A*). This mutant reported wildtype levels of enhancer activity in Tv4-neurons; therefore, we discounted all sequences 5′ of the pMad-binding site as essential (*Figure 1—figure supplement 1B,C*).

Within sequences 3′ of pMad-binding site in the remainder of the minimal 39 bp region, there are no consensus *GNCV* or *GTCT* motifs. However, three sequences deviate from the consensus by only a single nucleotide, including a perfectly conserved *GTAT* sequence spaced five nucleotides from the pMad site (yellow highlight in *Figure 1B*), and a conserved *GTTACA* spaced 12nt from the pMad motif, with partially overlapping *GTTA* and *TACA* sequences (orange highlight in *Figure 1B*).

### pMad and Medea are direct activators of *FMRFa*

The lack of a canonical Medea site, necessary for BMP-RE motif activity, led us to test if *Medea* is even required for *FMRFa* expression. In two different null *Medea* backgrounds, immunoreactivity to the FMRFa prepropeptide was entirely lost (*Figure 1D–F*). Anti-Eya staining revealed that all four Tv neurons were generated. We also examined EYFP reporter expression driven from the full length 445 bp *Tv^WT^-EYFP* enhancer, and from the concatemerized *BMP-RCE* and *HD-RE* reporters (*Berndt et al., 2015*). As expected, *Tv^WT^-EYFP* and *BMP-RCE-EYFP* expression were eliminated,

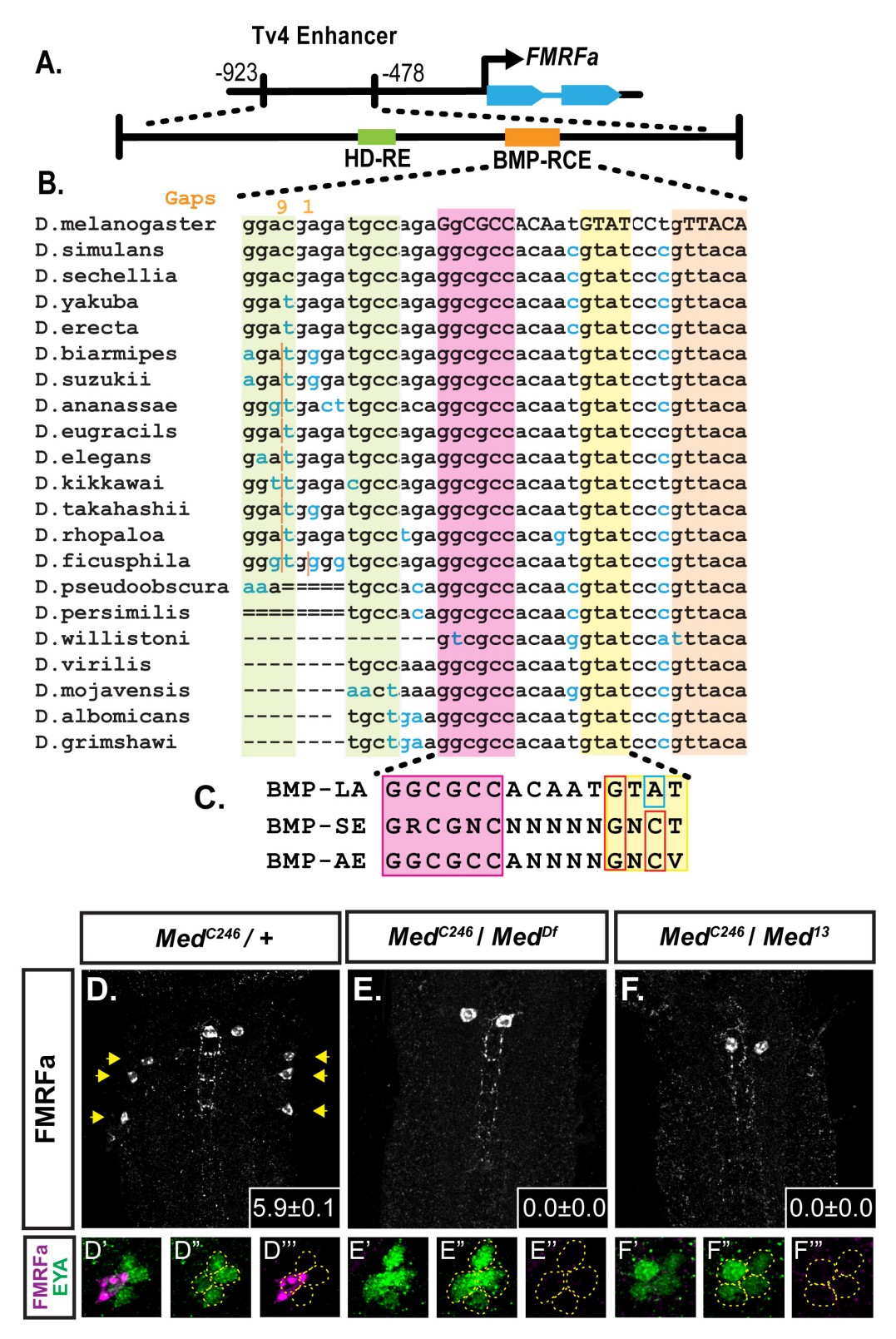

**Figure 1.** A novel BMP-Response element (BMP-RE) in the Tv4-neuron-specific enhancer of the *FMRFa* gene. The 445 bp Tv4-enhancer depicted in (**A**) contains two *cis*-elements critical for *FMRFa* activation, the homeodomain response element (HD-RE that recruits Apterous), and the BMP response element (BMP-RCE) that binds pMad and mediates the BMP-dependence of the Tv4-enhancer. (**B**) Output from the UCSC Browser shows sequence conservation through the BMP-RE across 21 *Drosophila* species. Capitalized letters are conserved across all species. Highlighted sequences include the

*Figure 1 continued on next page*

*Figure 1 continued*

putative Mad/Brinker binding site (magenta), two 5′ optimal *GNCV* Medea binding sequences with non-canonical spacing (green), and three 3′ sequences that deviate from a non-stringent *GNCN* sequence in one critical nucleotide (yellow, orange). The yellow *GTAT* motif is ideally spaced but has a C > A switch. The orange *GTTACA* contains two motifs (*GTTA* with a C > T switch, and *TACA* with a G > T switch). (C) Comparison of a putative *FMRFa* BMP-RE with the well-defined BMP-SE and BMP-AE motifs. (D-F) FMRFa immunoreactivity in Tv4-neurons is lost in *Medea* nulls (E,F) compared to controls (D). Insets show mean number of *FMRFa*-positive Tv4-neurons per VNC ± SD. (D′-F′′) Fluorophore splits of single Tv -clusters in A-C, showing Tv-neurons (circled) labeled by anti-Eya (green) and anti-FMRFa (magenta) expression in each genotype.

The online version of this article includes the following figure supplement(s) for figure 1:

**Figure supplement 1.** *FMRFa* expression in Tv4-neurons requires *Medea*.

while the BMP-insensitive *HD-RE-EYFP* exhibited expression comparable to controls (*Figure 1—figure supplement 1D–I*). Thus, *Medea* is selectively required for the BMP input that activates *FMRFa* expression.

The apparent dichotomy of *Mad* and *Medea*-dependence in the absence of a canonical BMP-RE motif led us to test whether *brinker* (*brk*) de-repression accounts for BMP-dependent *FMRFa* expression (*Figure 2A*). In numerous *Drosophila* tissues outside the nervous system, *brk*-mediated de-repression results in BMP-dependent gene expression (*Hamaratoglu et al., 2014*). In the absence of BMP-signaling, Brk binds *GGCGYY* motifs to default repress many genes that are activated upon the onset of BMP signaling (*Zhang et al., 2001*). In the presence on BMP signaling, pMad, Medea, and Shn bind BMP-SE motifs to repress *brk* (*Pyrowolakis et al., 2004*; *Marty et al., 2000*; *Torres-Vazquez et al., 2000*; *Müller et al., 2003*), lowering its expression and resulting in the de-repression of BMP gene targets (*Pyrowolakis et al., 2004*; *Jaźwińska et al., 1999*; *Campbell and Tomlinson, 1999*; *Winter and Campbell, 2004*; *Minami et al., 1999*; *Moser and Campbell, 2005*).

The BMP-responsive *GGCGCC* sequence in the *FMRFa* BMP-RCE matches this Brk consensus motif (*Zhang et al., 2001*). If a Brk de-repression model were correct for *FMRFa* expression (*Figure 2H*), BMP-induced pMad, Medea, and Schnurri (Shn) would be required to silence *brk*, resulting in *FMRFa* de-repression (*Marty et al., 2000*; *Müller et al., 2003*). Thus, we would expect that Brk expression would be increased in *wit* nulls, resulting in *FMRFa* repression. To test this model genetically, we first examined *brk*$^{XA}$/Y hemizygotes, but observed no change in either anti-FMRFa immunoreactivity or *Tv4*$^{WT}$-EYFP reporter expression (*Figure 2B,E*). Second, we tested *brk;;wit* double mutants to test if the absence of *FMRFa* in *wit* mutants is due to upregulation of the *brk* repressor. Contrary to this hypothesis, *FMRFa* expression was absent in *brk*/Y;;*wit*$^{A12}$/*wit*$^{B11}$ double mutants (*Figure 2D,E*), phenocopying *wit* mutants (*Figure 2C,E*).

However, this analysis did not rule out the possibility that *FMRFa* activation requires de-repression by *brk* as well as direct activation by pMad/Medea, as occurs at BMP-AE motifs regulating *dad* (*Weiss et al., 2010*). In such a model, the absence of *FMRFa* in *brk; ;wit* nulls may be due to a lack of activation by pMad/Medea. Therefore, we tested *FMRFa* expression in *shn* nulls. In this genotype, Brinker and pMad/Medea would all be expressed. In *shn* (*Allan and Thor, 2015*) null mutants, we found that FMRFa immunoreactivity was wildtype in late stage 17 embryos, the latest age testable in these mutants due to *shn* (*Allan and Thor, 2015*) late embryonic lethality (*Figure 2G*). Thus, *brk* de-repression cannot function as the primary mechanism for BMP-dependent *FMRFa* expression. We conclude that *FMRFa* is directly activated by pMad/Medea (*Figure 2I*), with no apparent involvement for *shn* and *brk*-mediated de-repression.

## The 15 bp FMRFa BMP-RE has reduced pMad/Medea binding relative to canonical BMP-REs

These data led us to test if the pMad/Medea complex is recruited by a non-canonical motif within the *FMRFa* BMP-RCE. We used an Electrophoretic Mobility Shift Assay (EMSA) to examine nucleotides essential for pMad/Medea binding. We activated BMP signaling in S2 cells, by transfecting S2 cells with FLAG::Mad, Myc::Medea, and activated BMP type I receptor Thickveins (Tkv$^{QD}$) (*Gao et al., 2005*). From these cells, we obtained total cell lysates to perform EMSA tests on IRDye700-tagged DNA oligonucleotides containing a 27 bp BMP-RE sequence. In *Figure 3*, we show that co-transfection of all three plasmids was required for a band shift of the *FMRFa* BMP-RCE probe by BMP-activated S2 cell lysates. Addition of either an anti-Myc or anti-FLAG IgG to the lysate

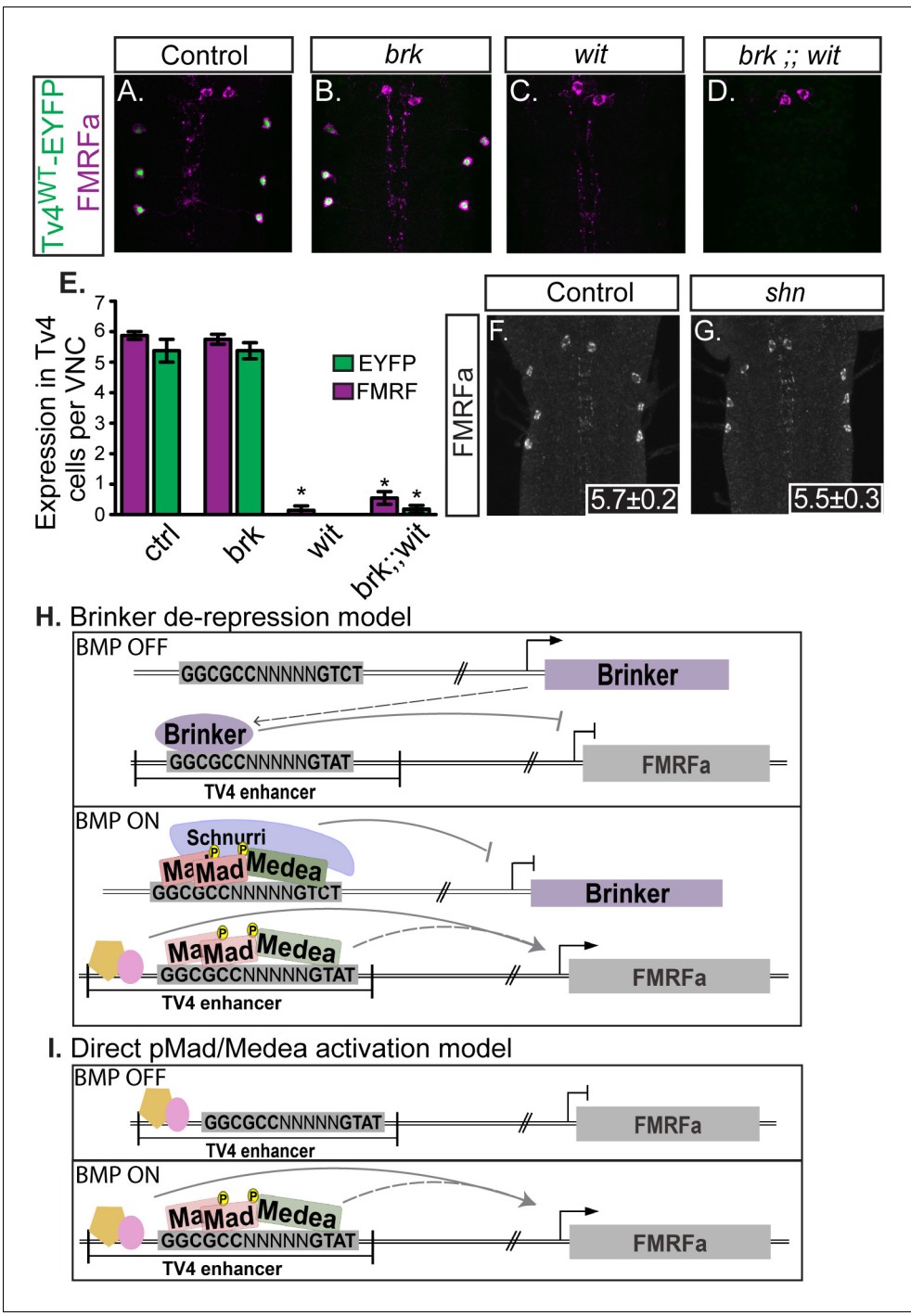

**Figure 2.** Neither *brinker* nor *schnurri* are required for BMP-dependent *FMRFa*. (A,B) Expression of the BMP-dependent FMRFa prepropeptide and the *Tv4^{WT}-EYFP* reporter were not affected in *brk* mutants. (C,D) Loss of FMRFa prepropeptide and *Tv4^{WT}-EYFP* in *wit* nulls was not rescued by the loss of *brk* in the double mutant of *brk* and *wit*. Thus, *FMRFa* is not lost in *wit* mutants due to the de-repression of *brk*. (E) Quantification of data in A-D (n = 7–12 animals per group, *p<0.01 compared to controls using One-way ANOVA with Tukey HSD *post-hoc*). (F, G) No change in the number of FMRFa-positive Tv4-neurons was observed between control and *shn* nulls in late stage 17 embryos. Mean ± SD number of FMRFa-positive Tv4-neurons per VNC shown in inset, n = 5 per genotype. (H) In a Brinker de-repression model, *brinker* would act as an *FMRFa* repressor by binding to the BMP-LA element (at the *GGCGCC* motif). When the BMP signaling pathway is active, the activated pMad/Medea complex would translocate to the nucleus and bind to the *brk* BMP-SE element, recruiting Schnurri, and silencing *brk* expression. This would allows expression of *FMRFa*, activated by other transcription factors and/or direct

*Figure 2 continued on next page*

*Figure 2 continued*

binding of a pMad/Medea complex. (I) In a direct pMad/Medea complex activation model, *FMRFa* is only expressed when an activated pMad/Medea complex binds to the BMP-LA sequence. Our work indicates that this latter model is likely correct, as neither *brk* nor *shn* manipulation modulates BMP-dependent *FMRFa* expression. Genotypes: Control (Tv^WT^-nEYFP). brk (brk^XA^/Y;;Tv^WT^-nEYFP/Tv^WT^-nEYFP). wit (Tv^WT^-nEYFP,wit^A12^/Tv^WT^-nEYFP, wit^B11^). brk;;wit (brk^XA^/Y;; Tv^WT^-nEYFP,wit^A12^/Tv^WT^-nEYFP,wit^B11^). shn control in F (w;shn ^1^/+). shn null in G (shn^1^/ shn^1^).

super-shifted the band, whereas a control IgG had no effect. Thus, a BMP-activated pMad/Medea complex isolated from S2 cells bind and band shift the *FMRFa* BMP-RCE.

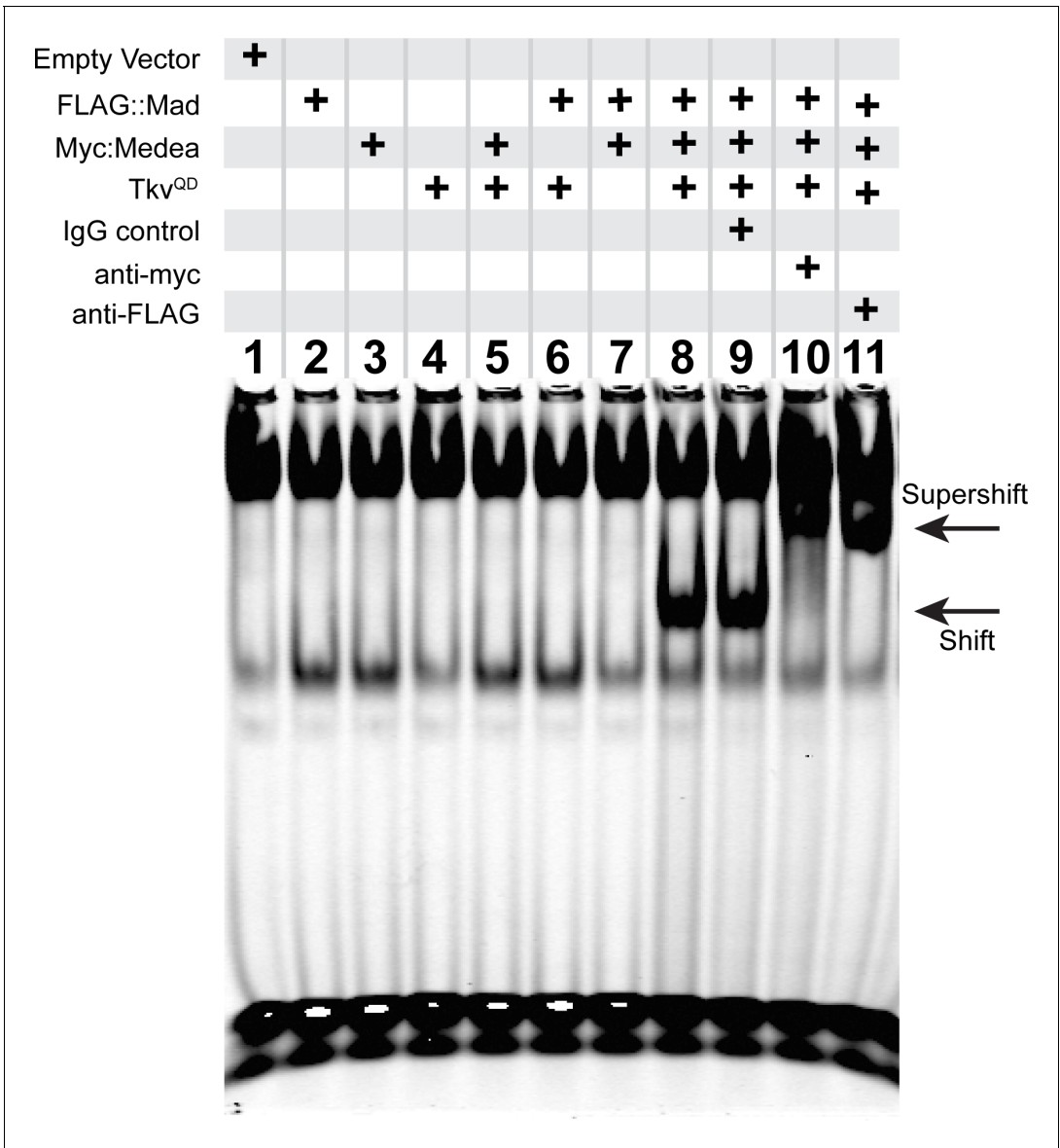

**Figure 3.** An activated pMad/Medea complex binds to the *FMRFa* BMP-RE. EMSA using IRDye700-tagged DNA oligonucleotides of the *FMRFa* BMP-RE incubated with S2 cell extracts transfected with FLAG::Mad and/or Myc::Medea, and/or activated BMP-receptor Tkv^QD^. Lanes 2–7 showed no specific band shift that differed from empty vector transfected cells (lane 1). Co-transfection of FLAG::Mad, Myc::Medea and Tkv^QD^ together generated a strong band shift (lane 8). Addition of antibodies to Myc (lane 10) or FLAG (lane 11) caused a super-shifted band that was not seen upon addition an IgG antibody control (lane 9). Thus, Mad, Medea, and activated Tkv receptor were capable of generating a band-shift of the *FMRFa* BMP-RE.

Next, we tested which sequences within the BMP-RCE are required for pMad/Medea recruitment. We generated an IRDye-700 tagged wildtype *FMRFa* BMP-RCE probe comprising a reduced 27 bp *FMRFa* BMP-RCE (*AGAGGCGCCACAATGTATCCCGTTACA*), necessary for appropriate in vivo expression (*Figure 1—figure supplement 1A*). We first tested candidate pMad and Medea binding sites by generating untagged probes with mutant sequence to test their ability to outcompete the wildtype tagged BMP-RE probe when pre-incubated at 10× or 100× excess. We confirmed that mutation of the pMad-binding sequence *GGCGCC* eliminated binding of the activated pMad/ Medea complex (*Figure 4A*), as previously demonstrated (*Berndt et al., 2015*). Next, we tested candidate Medea-binding sequences by mutating the *GTAT* sequence (to *TTAT*) and the sequence *CCCG* (to *AATT*). We found that only mutation of the *GTAT* sequence reduced binding of the activated pMad/Medea complex (*Figure 4B*). This indicated that a 15 bp *GGCGCC(N5)GTAT* motif likely represents the pMad/Medea-recruitment motif for the *FMRFa* BMP-RCE. We further corroborated these data by comparing the ability of the pMad/Medea complex to bind IRDye-700-tagged BMP-RE oligonucleotides that were either wildtype or contained GTAT mutations predicted to abrogate Medea binding (*ACTA* or *TTAT*). Our results demonstrated that pMad/Medea had reduced binding to the mutant sequences (*Figure 4—figure supplement 1*).

We wished to define the sequence requirements for this novel 15 bp BMP-Response Element (BMP-RE). Therefore, we examined band shifts of BMP-activated S2 cell lysates pre-incubated with tagged wildtype BMP-RE DNA oligonucleotides and a 100× stoichiometric excess of cold mutants in which a single nucleotide through the 15 bp motif was substitution mutagenized (*Figure 4C–E*). Mutagenesis of the entire *GGCGCC* sequence, or any single nucleotide therein, severely reduced binding of pMad/Medea, as shown by an inability to reduce the band shift of the tagged BMP-RE probe (*Figure 4C*). In addition, mutation of the entire *GTAT* sequence, or of any nucleotide except the *A*, greatly reduced the binding of pMad/Medea, as shown by retention of a strong band shift of the tagged BMP-RE probe (*Figure 4E*). By contrast, mutation of any nucleotide within the linker sequence *ACAAT* only minimally reduced pMad/Medea binding and the band shift was only minimally retained (*Figure 4D*). We conclude that a minimal 15 bp element of *GGCGCCacaatGTaT* is essential for pMad/Medea recruitment in vitro (with capitalized letters being essential).

It is intriguing that the position 14 *A* nucleotide in the putative *GTAT* Medea binding motif is the only nucleotide not required for pMad/Medea recruitment to this motif in vitro, as a *C* in this position was shown to be important for pMad/Medea recruitment to BMP-AE and BMP-SE motifs (*Pyrowolakis et al., 2004*; *Weiss et al., 2010*). Due to this loss of a key pMad/Medea recruitment nucleotide, we postulate that this position 14 *C > A* switch reduces affinity by introducing a nucleotide that fails to contribute to binding, resulting in the observed reduction of pMad/Medea recruitment.

Hereafter, we term the *FMRFa* BMP-RE motif as a BMP-Low Affinity Activation motif (BMP-LA). We wished to determine if the *C > A* switch that distinguishes the BMP-LA from the BMP-AE/SE motifs has an impact on BMP-activated pMad/Medea recruitment. To examine this, we tested whether modifying the BMP-LA Medea site to a BMP-AE-like or BMP-SE-like sequence indeed increases its ability to bind pMad/Medea. We generated a series of tagged and untagged DNA oligonucleotides comprising 27 bp of the BMP-LA sequence that was either wildtype at the Medea site, *GTAT,* or mutated at this site to *GACG* (BMP-AE-like) or *GTCT* (BMP-SE-like). By competition EMSA assays, we tested the relative ability of these untagged DNA oligonucleotides to compete for pMad/Medea binding when at 1×, 5×, and 10× stoichiometric ratios relative to the tagged wildtype probe (*Figure 4F–G*). We found that a wildtype untagged BMP-LA sequence reduced but did not eliminate the band shift generated by its tagged counterpart by 10×. In contrast, both the AE-like and the SE-like sequence mutants proved to be much stronger competitors, totally outcompeting the BMP-LA tagged probe by 5×. Moreover, at equimolar ratios, the untagged wildtype BMP-LA competitor did not substantially alter the tagged probe band shift; however, both AE-like and SE-like mutants reduced the band shift. These data indicate that the *GTAT* sequence displays low affinity pMad/Medea binding activity relative to the characterized BMP-AE and BMP-SE motifs.

## Conversion of the low affinity *FMRFa* BMP-RE to a high affinity BMP-AE sequence results in ectopic neuronal BMP-dependent activity in vivo

Collectively, our genetic and biochemical data indicated that the BMP-LA motif exhibits low affinity pMad/Medea recruitment. This raised the question as to why the BMP-LA motif is attenuated in this

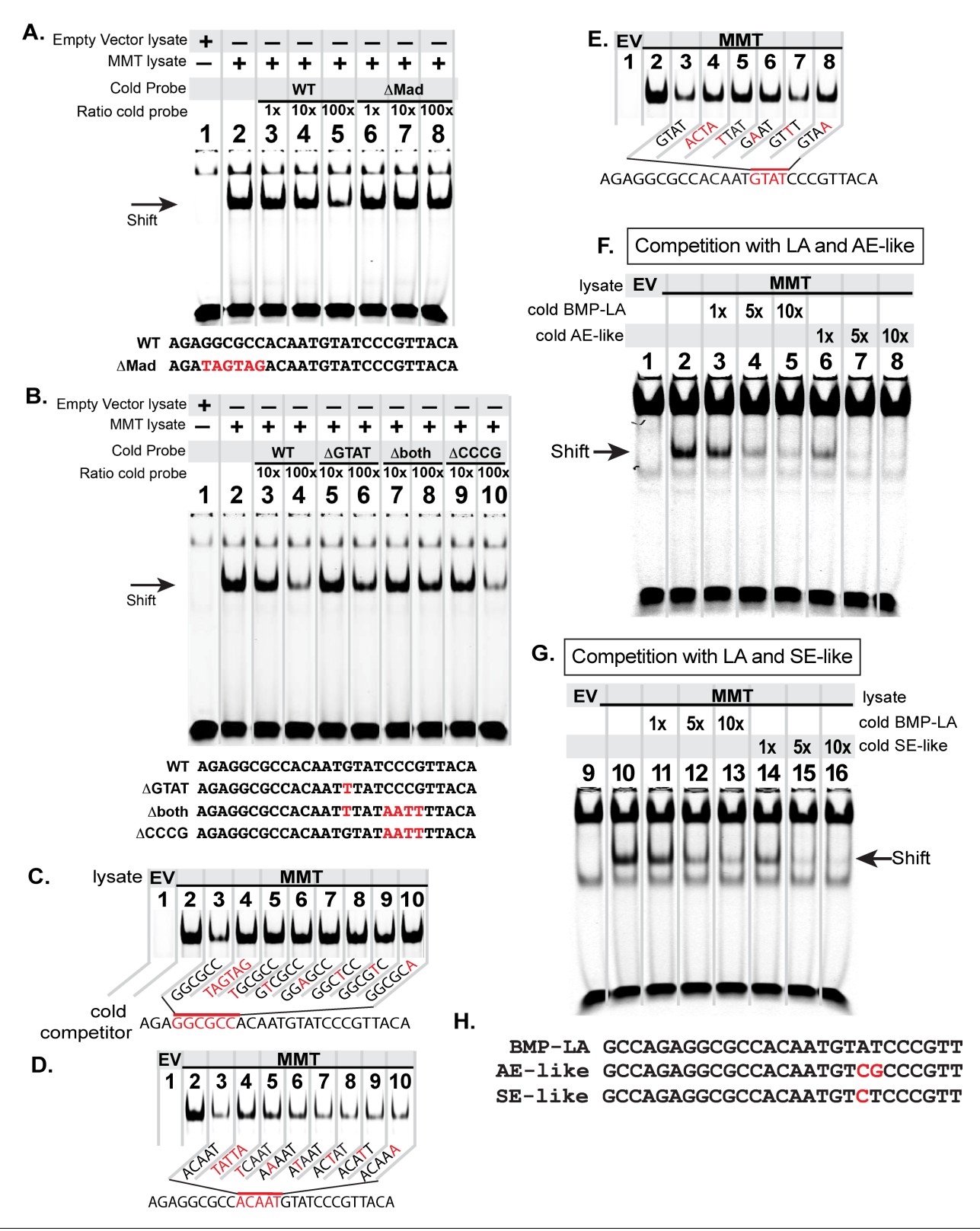

**Figure 4.** The *FMRFa* BMP-LA is required for pMad/Medea recruitment but at a lower affinity compared to BMP-AE and BMP-SE motifs. We performed EMSA gels in which we ran IRDye700-tagged *FMRFa* BMP-RE DNA oligonucleotides (of sequence *AGAGGCGCCACAATGTATCCCGTTACA*) pre-incubated with lysates from S2 cells transfected with either empty vectors (EV; lane 1) or FLAG::Mad, Myc::Medea and activated BMP-receptor, Tkv^QD (MMT lysate; lanes 2–8 or 2–10) (**A-B**). The MMT lysate generated a band shift indicative of pMad/Medea binding to the tagged probe (lane 2 in **A,B**). In lanes

*Figure 4 continued on next page*

*Figure 4 continued*

3–8 (**A**) or 3–10 (**B**), we ran MMT lysates pre-incubated with tagged probe and a stoichiometric excess of untagged (cold) DNA oligonucleotides with either wildtype or mutated sequence (shown below each gel). Loss of a band shift indicates that the untagged probe is capable of binding activated pMad/Medea. (**A**) We pre-incubated with untagged (cold) competitors of wildtype (WT) sequence or *GGCGCC >TAGTAG* mutated sequence (Δ*Mad*) in 1×, 10×, 100× excess. At 100× excess, competition by the WT cold probe reduced the band shift (lane 5). In contrast, the mutated cold probe failed to reduce the band shift at 100× excess (lane 8). (**B**) We pre-incubated with 10x and 100x excess of cold probes of wildtype (WT) or mutated sequences, including of G and C nucleotides within candidate Medea-binding sequences *GTAT* (>*TTAT*, termed Δ*GTAT*), *CCCG* (>*AATT*, termed Δ*CCCG*), and a mutant to both of these sequences (Δ*both*). At 100× excess, the wildtype (lane 4) and Δ*CCCG* (lane 10) greatly reduced the band shift to the same extent; thus, the *CCCG* sequence does not contribute to pMad/Medea binding. By contrast, the Δ*GTAT* (lane 6) and Δ*both* (lane 8) cold probes only partially reduced the band shift, indicating that their ability to bind pMad/Medea was compromised. These data suggest that the minimal BMP-RE comprises a *GGCGCC(N₅)GTAT* sequence. (**C-D**) The MMT lysate generated a band shift indicative of pMad/Medea binding to the tagged probe (lane 2). We added untagged (cold) BMP-RE DNA oligonucleotides at 100× stoichiometric excess with mutations in the sequences shown in red. (**C**) Nucleotide mutations are shown as red within the pMad-binding site. (**D**) Nucleotide mutations are shown as red within the linker region. (**E**) Nucleotide mutations are shown as red within the Medea-binding site. Examining the ability of each cold competitor to compete with tagged *FMRFa* BMP-RE, we reveal a necessary BMP-RE sequence of *GGCGGGacaatGTaT*, where capitalized nucleotides are found most necessary for pMad/Medea recruitment. (**F,G**) In these EMSA, we additionally transfected S2 cell extracts with a 1×, 5×, and 10× stoichiometric excess of untagged competitors (sequences shown in **H**). (**F**) Competition for the tagged BMP-LA by the untagged wildtype (BMP-LA) or the AE-like mutant. A 10× excess (lane 5) of untagged wildtype BMP-LA reduced but did not eliminate the band shift (lane 5). By contrast, the BMP-AE-like mutant totally out-competed the tagged BMP-LA at a 5× excess (lane 7 compared to lane 4) and significantly out-competed the tagged BMP-LA even at equimolar ratio (lane 6 compared to lane 3). (**G**) Competition for the tagged BMP-LA by the untagged wildtype or the SE-like mutant. The SE-like mutant totally out-competed the tagged BMP-LA at a 5× excess (lane 15 compared to lane 12) and significantly out-competed the tagged BMP-LA even at equimolar ratio (lane 14 compared to lane 11). The online version of this article includes the following figure supplement(s) for figure 4:

**Figure supplement 1.** Reduced pMad/Medea recruitment upon GTAT sequence mutation in the *FMRFa* BMP-RE.

manner, yet conserved across all *Drosophila* species. To address this, we tested for functional relevance for this motif's low affinity.

First, we mutated the 445 bp Tv4-enhancer (within the *Tv4-nEGFP* nuclear-localized reporter) at the *GTAT* sequence into *ACTA* (*Tv4^mGTAT>ACTA^-nEGFP*) (sequence shown in *Supplementary file 1a*), in order to eliminate Medea recruitment. We found that the *Tv4^mGTAT>ACTA^-nEGFP* reporter exhibited a total loss of expression in Tv4-neurons, and no ectopic expression was detected in other neurons (*Figure 5A,B*). We conclude that the *GTAT* sequence is essential. Second, we tested the effect of converting the *GTAT* motif of BMP-LA to an optimal BMP-AE-like sequence (*GTAT > GACG*), within the *Tv4-nEGFP* reporter (*Tv4^mGTAT>GACG^-nEGFP*) (sequence shown in *Supplementary file 1a*). This conversion eliminated Tv4-enhancer activity in Tv4-neurons (*Figure 5A,C*) but led to strong ectopic reporter activity in other neurons of the VNC (*Figure 5D*). We noted that all EGFP positive cells were also anti-pMad immunoreactive (*Figure 5D'–D'''*), suggesting that we had indeed created a high affinity BMP-RE that recruits pMad/Medea to generate BMP-responsive Tv4-enhancer activity in an expanded population of neurons. To confirm this, we tested the *wit*-dependence of this ectopic reporter activity, by examining the expression of the mutant BMP-AE-like Tv4-enhancer reporter (*Tv4^mGTAT>GACG^-nEGFP*) in a *wit* mutant background (*Figure 5D,E*). As expected, this eliminated all reporter activity, demonstrating that the low-to-high affinity conversion created a functional BMP-AE that generated BMP-dependent reporter activity in other BMP-activated neurons of the VNC.

These data show that the low affinity of BMP-LA is essential for the spatially restricted activity of this *cis*-element to Tv4-neurons.

## Identification of additional functional BMP-responsive BMP-LA motifs through the genome

Following the identification of the novel BMP-LA motif regulating *FMRFa* expression in Tv4-neurons, we examined if this *cis*-regulatory motif was unique to *FMRFa* regulation, or functions to confer BMP-dependence to other *cis*-regulatory regions. To this end, we identified 178 BMP-LA motifs in the *D. melanogaster* genome using the motif discovery tool HOMER (v4.10) (*Heinz et al., 2010*). These were filtered for sequence conservation across 24 sequenced *Drosophila* species using Phast-Cons scores (*Siepel et al., 2005*), reducing the list to 128 BMP-LA with an average PhastCons score over 0.55. Of these 128 BMP-LA, 103 were highly conserved with a score of over 0.9 (*Supplementary file 1b*).

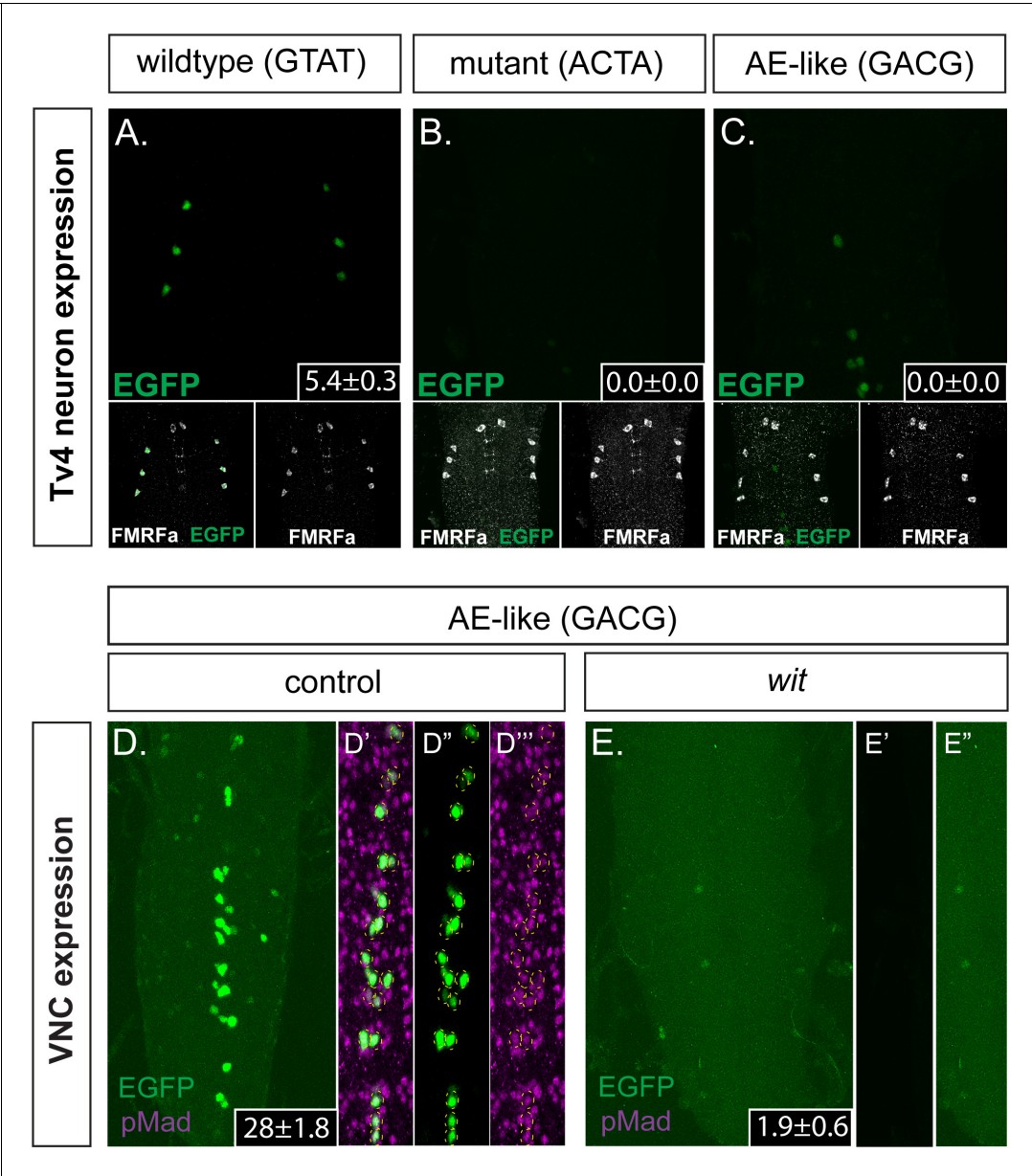

**Figure 5.** The *FMRFa* BMP-LA motif has a necessary but low affinity Medea binding site that specifies selective neuronal subtype activity. (**A–C**) Conversion of the Medea-binding *GTAT* site in the wildtype 445 bp Tv4-neuron-specific *FMRFa* enhancer to a mutant version, *ACTA* (that reduces pMad/Medea recruitment; termed *Tv4 ^mGTAT>ACTA^–nEGFP*), resulted in a complete loss of reporter gene expression in Tv4-neurons (**B**). Conversion of the Medea-binding *GTAT* site in the wildtype 445 bp Tv4-neuron-specific *FMRFa* enhancer to an optimal BMP-AE-like sequence (*GACG;* termed *Tv4 ^mGTAT>GACG^–nEGFP*) also resulted in a total loss of reporter expression in Tv4-neurons (**C**). Numbers in insets indicate the mean ± SD number of EGFP-positive Tv4-neurons per VNC, out of the possible six Tv neurons. (**D,E**) The *Tv4^mGTAT>GACG^–nEGFP* reporter generated strong ectopic reporter activity in VNC midline cells (**D**) that is lost in the absence of neuronal BMP signaling (E; in *wit* mutants, *wit^A12^/wit^B11^*). Full z-projections though the whole VNC are shown. Numbers in insets indicate the mean ± SD number of EGFP-positive neurons per VNC. (**D'–D'''**) Images of the midline ectopic EGFP expression generated from *Tv4^mGTAT>GACG^–nEGFP*. EGFP expression (green) was exclusively expressed in pMad-immunoreactive cells (magenta); all cells are yellow circled.

We selected 24 of the BMP-LA motifs for functional testing *in vivo*. Of the 24 prioritized motifs, 20 were highly conserved (average PhastCons score >0.9) and chosen in order to optimize the chance of characterizing functional BMP-RE's. Additionally, four motifs with lower scores 0.75–0.55 were tested for functionality (*Table 1*, *Supplementary file 1c*). We examined reporter activity driven from these genomic fragments in late third instar larvae (L3). Of the 24 reporters, five showed no

**Table 1.** Summary table of expression pattern and *wit*-responsiveness for BMP-LA containing DNA fragments tested in vivo.
The first column indicates the name for each of the cloned BMP-LA containing DNA fragments; these were sorted based on intensity and pattern of reporter expression. The second column provides information on reporter expression in the VNC, (more plus signs indicate higher intensity), and the final column provides details on expression pattern. The third and fourth columns indicate fragments that exhibited pMad and reporter co-expression, as well as the ones that were shown to be *wit*-responsive. Bolded letters indicate the enhancer fragments that were further tested for *wit*-responsiveness. The expression pattern was assessed in wandering third instar larvae.

| DNA fragment | VNC expression | Reporter/pMad stain overlap | *Wit* responsive | VNC expression details |
|---|---|---|---|---|
| CM5 | +++ | √ | √ | neurons and glia |
| CM1 | +++ | √ | √ | medial and lateral neurons |
| CM4 | +++ | √ | √ | medial and lateral neurons |
| CM3 | +++ | √ | √ | medial and lateral neurons |
| CM2 | ++ | √ | √ | medial and lateral neurons |
| CM7 | + | √ | √ | sparse |
| CM6 | + | √ | √ | sparse |
| CM8 | +++ | √ | - | medial neurons |
| CM9 | ++ | √ | - | lateral neurons |
| CM10 | + | √ | - | sparse |
| CM11 | ++ | - | - | neurons and glia |
| CM12 | + | - | - | sparse |
| CM13 | + | - | - | sparse |
| CM14 | + | - | - | sparse |
| CM15 | + | - | - | sparse |
| CM16 | + | - | - | sparse |
| CM17 | + | - | - | low intensity |
| CM18 | + | - | - | low intensity |
| CM19 | + | - | - | low intensity |
| CM20 | - | - | - | none |
| CM21 | - | - | - | none |
| CM22 | - | - | - | none |
| CM23 | - | - | - | none |
| CM24 | - | - | - | none |

expression in the ventral nerve cord (VNC) (*Figure 6—figure supplement 1A*), while the remaining 19 reporters exhibited low to high reporter activity in the VNC (*Table 1*). Of these active reporters, 10 exhibited expression in subsets of pMad-positive cells in the VNC, and also numerous pMad-negative glia and neurons (*Table 1*). The remaining nine active reporters were restricted to pMad-negative glia and neurons (*Figure 6—figure supplement 1B*). We tested the BMP-responsiveness of the 10 reporters expressed in pMad-positive cells, by placing them in a *wit* mutant background. Out of those 10 reporters, seven showed reduced reporter expression (*Figure 6B*), including two (CM5 and CM7) with motifs of lower PhastCons score. Three reporters showed no significant change (*Figure 6A*, *Figure 6—figure supplement 1B*). Quantification of reporter expression in the VNC of controls and their corresponding *wit* mutants revealed a loss of up to 88% of reporter-expressing cells in certain genotypes (*Figure 6—figure supplement 2B*).

Next, we tested whether the activity of the identified *wit*-responsive fragments was dependent on the BMP-LA motif within these genomic fragments. We selected four *wit*-responsive fragments (CM1, CM2, CM3, and CM7) and one non-*wit* responsive fragment (CM9; *Figure 7A*). We introduced nucleotide substitution mutations into the pMad-binding site of the LA

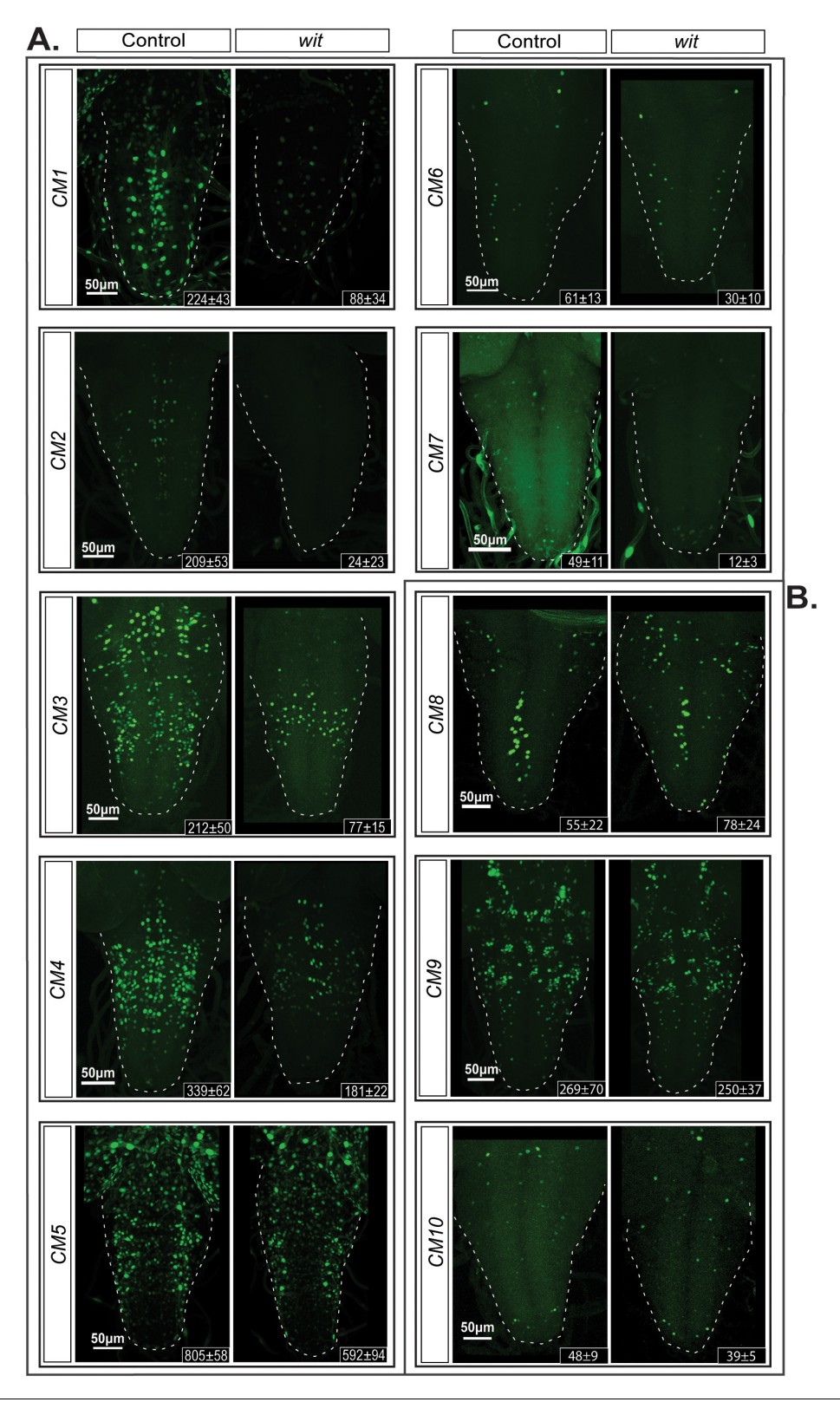

**Figure 6.** Identification of additional *wit*-responsive genomic fragments containing the BMP-LA motif. We identified 10 genomic fragment reporters that exhibited expression in subsets of pMad-positive cells in the VNC and tested their *wit*-responsiveness. (**A**) EGFP reporter patterns of three genomic fragments that exhibit no *wit*-responsive loss of reporter expression in *wit* mutants ($wit^{A12}/wit^{B11}$) compared to controls ($wit^{A12}/+$) in late third

*Figure 6 continued on next page*

*Figure 6 continued*

Instar larval VNCs. (**B**) Nuclear EGFP expression patterns driven from seven genomic fragments containing conserved BMP-LAs that were down-regulated in *wit* mutants (*wit^{A12}/wit^{B11}*), compared to controls (*wit^{A12}/+*) in late third Instar larval VNCs. The observed down-regulation ranged from a near-total loss of all neuronal expression to loss of expression in a subset of neurons. Full z-projections though the whole VNC are shown. *Genotypes:* All control lines shown here were heterozygous (*w;;CM#/+*); *wit* mutants (*w;;CM#,wit^{A12}/wit^{B11}*). The online version of this article includes the following figure supplement(s) for figure 6:

**Figure supplement 1.** BMP-LA containing reporter fragments with no reporter and pMad co-expression.
**Figure supplement 2.** Quantification of reporter expressing cells reveals significant downregulation in *wit* mutants or pMad-binding site mutants compared to their respective controls.

---

(*GGCGCC > TGATGA*). In all four *wit*-responsive fragments tested, there was a significant loss of reporter expressing cells (*Figure 7B*). Finally, having established the necessity of *Medea* for BMP-dependent activity of the *FMRFa* BMP-RE, we tested the requirement for *Medea* in reporter expression these same BMP-dependent CM1, CM2, CM3, and CM7 reporters. In *Medea* mutant third instar (L3) larvae, reporter expression was reduced in the same pattern as observed in a *wit* mutant background (*Figure 7—figure supplement 1*).

We conclude that the BMP-LA is utilized by numerous *cis*-regulatory regions in order to generate BMP-dependent enhancer activity in neurons.

## Discussion

Retrograde BMP signaling, and BMP responsive DNA motifs in target gene enhancers, have been viewed as functionally equivalent across efferent neuron populations. Therefore, the differential expression of BMP target genes across efferent neuron populations has been attributed to the activity of subtype-specific transcription factors (*Veverytsa and Allan, 2011*; *Miguel-Aliaga et al., 2008*; *Allan et al., 2005*). Here we describe our identification of a low affinity BMP-responsive DNA motif that appears to instructively contribute to differential specification of BMP target genes across efferent neuron subtypes.

Our previous work had identified a 39 bp BMP responsive *cis*-element within the Tv4-enhancer of *FMRFa* that is absolutely required for Tv4-enhancer activity, and encodes sufficient information for selective activity in Tv4-neurons (*Berndt et al., 2015*). Here we reveal that pMad and Medea are recruited to this *cis*-element by an essential low affinity motif (the BMP-LA). Converting this motif to a high affinity BMP-AE motif led to Tv4-enhancer activity in a broader set of efferent neurons, as would be expected from BMP-AE's generic responsiveness across many efferent neurons and *Drosophila* tissues (*Vuilleumier et al., 2019*; *Chayengia et al., 2019*). These results provide evidence that low affinity BMP responsive motifs instructively restrict BMP target genes within subsets of efferent neuron.

Low affinity motifs serve widespread roles in restricting gene expression in space and time (*Crocker et al., 2015*). They function to reduce enhancer activity in order to prevent ectopic expansion of enhancer activity (*Wharton et al., 2004*; *Farley et al., 2015*; *Jiang and Levine, 1993*). They are also required to generate enhancer activity with the appropriate expression domain; for example, Hedgehog regulation of numerous wing imaginal disc and embryonic enhancers requires Ci transcription factor binding to low affinity motifs; increasing their affinity abolished enhancer activity (*Ramos and Barolo, 2013*). Also, clustered low affinity motifs have been shown to improve discrimination between transcription factors with similar binding preferences, and to locally concentrate transcription factors to ensure robust target gene expression (*Crocker et al., 2015*).

How does the BMP-LA confer Tv4-neuron specificity? We propose that the BMP-LA does not singularly specify Tv4 neuron activity, but rather imposes a requirement for pMad/Medea cooperative interactions with subtype-specific transcription factors bound to adjacent motifs for combinatorial activation of *FMRFa* expression. Four primary lines of evidence inform our conclusion. First, lowering or ectopically hyperactivating BMP signaling in other neurons failed to trigger ectopic Tv4-enhancer or *FMRFa* expression, without also co-misexpressing Tv4-specific transcription factors (*Allan et al., 2003*; *Allan et al., 2005*; *Miguel-Aliaga et al., 2004*; *Eade et al., 2012*; *Berndt et al., 2015*). This rules out a model whereby BMP-LA's lower affinity is tuned for a specific level of BMP signaling only

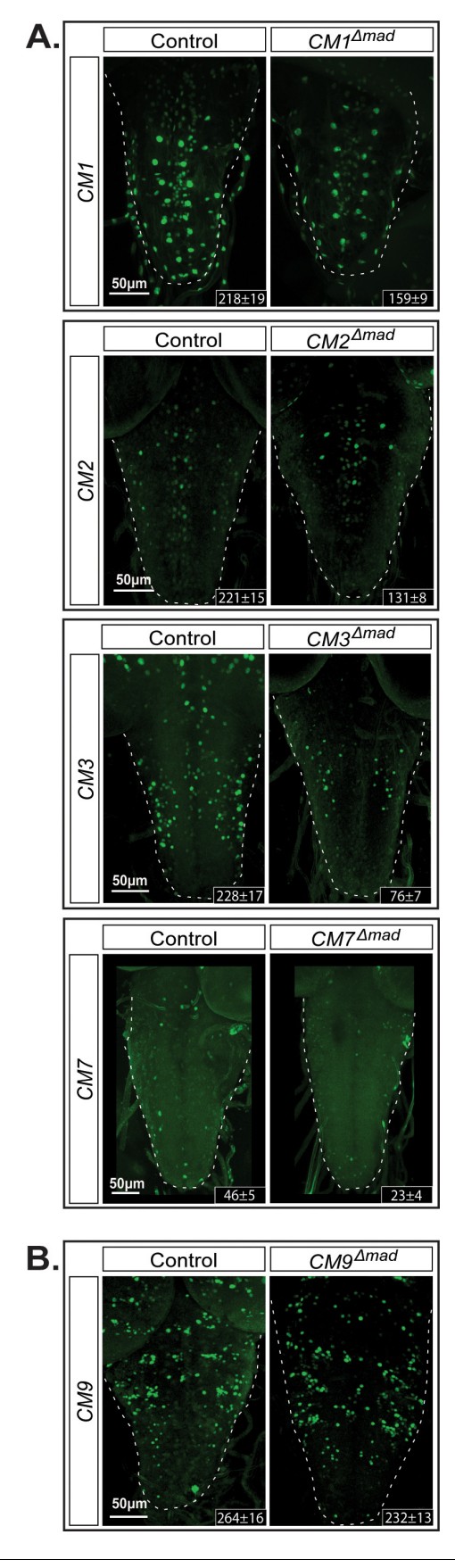

**A.**

| CM1 | Control | CM1^Δmad |
|---|---|---|
| | 50μm 218±19 | 159±9 |

| CM2 | Control | CM2^Δmad |
|---|---|---|
| | 50μm 221±15 | 131±8 |

| CM3 | Control | CM3^Δmad |
|---|---|---|
| | 50μm 228±17 | 76±7 |

| CM7 | Control | CM7^Δmad |
|---|---|---|
| | 50μm 46±5 | 23±4 |

**B.**

| CM9 | Control | CM9^Δmad |
|---|---|---|
| | 50μm 264±16 | 232±13 |

observed in Tv4-neurons. Second, the minimal BMP-responsive, Tv4-specific *cis*-element within the Tv4-enhancer cannot dispense with a highly conserved sequence 3' of the BMP-LA. This sequence likely contains binding sites for other transcription factors that synergize with pMad/Medea to specify *FMRFa* expression in Tv4-neurons. These transcription factor(s) remain unidentified at this time. Third, BMP-LA motifs in other genomic regions confer BMP-dependent enhancer activity across other efferent neurons, suggesting that the BMP-LA imposes a requirement for pMad/Medea synergistic interactions with other transcription factors that combinatorially specifies neuronal subtype expression. Fourth, converting the BMP-LA to a BMP-AE resulted in a loss of reporter expression within its appropriate domain (Tv4-neurons), but an inappropriate expansion of BMP-responsive activation into other efferent neurons. This strongly suggests that the BMP-LA imposes transcription factor interactions that the BMP-AE is unable to replicate in Tv4-neurons; yet are unnecessary for generic BMP-AE activity in other efferent neurons. These specific interactions may occur between transcriptional regulators that require a specific pMad/Medea conformation that is imposed by the BMP-LA sequence. Precedent for this comes from analysis of Schnurri binding to pMad and Medea bound at the BMP-SE, which requires a pMad/Medea conformation that is imposed by a precise 5nt linker between the Mad and Medea binding motifs and also a terminal T at position 15 of the BMP-SE motif (*Gao et al., 2005*; *Pyrowolakis et al., 2004*; *Müller et al., 2003*). No such sequence constraints appear to be required for pMad/Medea activity at the BMP-AE (*Weiss et al., 2010*; *Chayengia et al., 2019*). Alternatively, but not mutually exclusively, these specific interactions may occur between other transcription factors bound to adjacent DNA motifs and a pMad/Medea conformation imposed by the BMP-LA sequence. Precedent for such a model comes from evidence that conversion of a low affinity Pax6 site in the chicken DC5 enhancer to a high-affinity site disrupted the formation of an activation competent Sox2-Pax6 complex, and decreased enhancer activity (*Kamachi et al., 2001*; *Uchikawa et al., 2003*). Similarly, single nucleotide differences in NF-κB binding sequences determine cofactor specificity for NF-κB dimers (*Leung et al., 2004*; *Pan et al., 2010*). A similar instructive role for low affinity motifs in generating exquisitely subtype-specific neuronal gene expression has been proposed in *Drosophila* (*Jafari and Alenius, 2015*;

**Figure 7.** The *GGCGCC* pMad-binding site is necessary for reporter expression in vivo. We introduced specific mutations into the pMad-binding site of the BMP-LA motif (*GGCGCC >TGATGA*) of four *wit*-responsive fragments (CM1, CM2, CM3, and CM7) and one non-*wit* responsive fragment (CM9) to verify whether reporter activity was dependent on pMad binding. (**A**) CM9 showed no significant loss of reporter expression in the $CM9^{\Delta mad}$ mutant compared to the control. (**B**) All four *wit*-responsive fragments exhibited a significant loss of reporter expressing cells; however, this loss was less pronounced than the loss in *wit* mutants, apart from CM3. Full z-projections though the whole VNC are shown. *Genotypes*: All control and pMad-binding site mutant lines examined here were heterozygous (*w;;CM#/+*).

The online version of this article includes the following figure supplement(s) for figure 7:

**Figure supplement 1.** Medea is necessary for BMP-dependent activity of four *wit*-responsive BMP-LAs.

*González et al., 2019*). The exclusive expression of odorant receptor genes in specific olfactory sensory neurons is encoded by arrays of low affinity binding sites for multiple transcription factors. The authors propose that these low affinity motifs provide the necessary environment for the weak and somewhat promiscuous binding for numerous transcription factors that ultimately establishes the cooperative interactions that stabilizes target gene activation in a single neuronal subset.

Our results add to growing evidence that BMP-dependent gene regulation can be profoundly altered by subtle nucleotide substitutions within the 4-nucleotide Medea-recruitment site of these response elements; a BMP-dependent activator (*GNCV*), a BMP-dependent repressor (*GNCT*), and a low affinity activator that generates restricted activation (*GTAT*). The sequence similarity yet diverse functional output of these three BMP-REs reveals the diversity of responses that can be generated from subtle sequence deviations from a core BMP-RE motif. This raises a challenge for computational approaches for BMP-RE motif discovery; a specific high affinity motif will not capture all functional motif instances, but a degenerate motif built from multiple BMP-RE motifs would increase background noise to unacceptably high rates. Moreover, our discovery here of a novel motif with predictive value only increases the promise that more BMP-REs with predictive value will be discovered. However, as demonstrated by numerous studies, accounting for low affinity sites has proven to be beneficial in identifying functional enhancers (*Zandvakili et al., 2018*; *Gurdziel et al., 2015*). Therefore, it remains important to identify novel BMP-responsive motifs with demonstrated predictive value in BMP-RE discovery. Regardless, considerable degeneracy has been found in the sequences of BMP-REs (*Ross and Hill, 2008*) and it is important to acknowledge that *in silico* BMP-RE discovery will always under-represent functional BMP-REs in any system.

We started this study by sub-mapping a minimal BMP-responsive, Tv4-specific *cis*-element to a short 27 bp region that contains the canonical *GGCGCC* pMad-binding site, but no canonical Medea-binding site. After dismissing the expected model that this motif would mediate Brk default repression and BMP-dependent de-repression, we defined a minimal 15 bp *GGCGCC(N5)GTAT* bipartite motif required for pMad/Medea recruitment. The low affinity of this motif arises from the *C > A* nucleotide switch at position 14; removing the *C* nucleotide required for pMad/Medea recruitment to BMP-AE and BMP-SE (*Pyrowolakis et al., 2004*; *Weiss et al., 2010*), for an *A* nucleotide that plays no role in recruitment (*Figure 4E*). Interestingly, this loss of recruitment activity at motif position 14 leads to increased nucleotide stringency at positions 2, 5, 12, and 13 for pMad/Medea recruitment (*Figure 4E*), which are either unnecessary or less necessary for recruitment to BMP-AE and BMP-SE (*Gao et al., 2005*; *Pyrowolakis et al., 2004*; *Weiss et al., 2010*). Thus, transversion of one core nucleotide requires compensatory pMad/Medea recruitment activity from other nucleotides within the overall 15 bp motif. This could be viewed as exerting an evolutionary pressure on these remaining nucleotides to not exhibit degeneracy across *Drosophila* species, which may provide an explanation for why the BMP-LA is so highly conserved. By contrast, other low affinity *cis*-elements exhibit considerable degeneracy, with the hypothesis being that a high affinity, optimal motif would be highly constrained in nucleotide composition, but there could be a variety of nucleotide substitutions that would result in lower affinity motif (*Crocker et al., 2015*; *Ramos and Barolo, 2013*; *Crocker et al., 2016*; *Farley et al., 2016*).

Overall, our results show that differential BMP-dependent gene expression in neuronal subtypes is not only conferred by the integration of pMad/Medea into a combinatorial transcription factor code. In addition, sequence variation of the BMP-RE that is bound by pMad/Medea provides additional critical information required to selectively *trans*-activate gene expression in specific neuronal

subtypes. Further work is required to identify transcription factors that bind the highly conserved sequences that flank the BMP-LA motif in the *FMRFa* Tv4-enhancer, or interact directly with the BMP-LA motif itself. Once such factors are identified, we will be able to directly assess the contribution of pMad/Medea affinity and specific transcription factors to the BMP-LA in order to form subtype-specific, activation-competent complexes.

## Materials and methods

### Fly genetics

Strains used: *Med^{C246}* (*McCabe et al., 2004*); *Med^{13}* (*Hudson et al., 1998*); *Med^{Df}* (*Df(3R)ED6361*) (*Ryder et al., 2007*); *wit^{A12}* and *wit^{B11}* (*Aberle et al., 2002*); *brk^{XA}* (*Campbell and Tomlinson, 1999*); *shn^1* ( *Grieder et al., 1995*). Mutants were kept over *CyO, Act-GFP TM3, Ser, Act-GFP* (*Reichhart and Ferrandon, 1998*), *CyO, twiGAL4, UAS-2xEGFP*, or *TM3, Sb, Ser, twiGAL4, UAS-2xEGFP* (*Halfon, 2002*). *w^{1118}* was used as the control genotype. Flies were maintained at 25°C, 70% humidity.

### Reporter transgene construction

*Tv^{WT}-EYFP*, *BMP-RE-EYFP*, *HD-RE-EYFP* were generated previously (*Berndt et al., 2015*). *Drosophila* transformations were performed by Rainbow Transgenic Flies, Inc (Camarillo, CA). Empty pThunderbird EGFP vector was generated from Tv-nEYFP and from sequence within pHstinger (*Berndt et al., 2015*; *Barolo et al., 2000*). *Tv4-nEYFP* was digested with AscI and SpeI. The multiple cloning site (MCS), HSP70 promoter, EGFP coding sequence, Tra nuclear localization signal, and SV40-polyA sequences from pHstinger (*Barolo et al., 2000*) were liberated with AscI and SpeI and ligated into the cut *Tv-nEYFP* backbone. The Tv4-enhancer was PCR-amplified from Oregon-R with XbaI and EcoRI adaptors, restriction digested and ligated into XbaI/EcoRI digested empty Tv4-nEYFP. SOE PCR generated nucleotide substitution and deletion mutants were inserted similarly. Summary of all mutations and concatemerization sequences in *Supplementary file 1a*. pMad-binding site mutagenesis was performed by Q5 Site-Directed Mutagenesis Kit (New England Biolabs, Ipswich, MA), using primers designed to introduce specific base pair substitutions to the Mad binding site (*GCCGGC > tgatga*), according to manufacturer's protocols. All constructs were verified by sequencing before the generation of transgenic fly lines. Fly transformations were performed by Rainbow Transgenic Flies, Inc (Camarillo, CA). All transgenic reporters were integrated into *attP2* (*Groth et al., 2004*).

### Immunochemistry

Immunochemistry was performed as previously described (*Eade and Allan, 2009*; *Berndt et al., 2015*). Primary antibodies: Rabbit α-FMRFa C-terminal peptide (1:1000, a gift from S. Thor) *Baumgardt et al., 2007*; Mouse α-Eya (1:100; MAb clone 10H6 DSHB; Iowa University, IA); Rabbit α-pMad (1:100, 41D10, Cell Signaling Technology, Danvers, MA), Chicken α-GFP antibody (1:1000, ab13970, Abcam, Ontario, Canada). Donkey anti-Rabbit and anti-Mouse conjugated to DyLight 488, Cy3, Cy5 (1:100, Jackson ImmunoResearch, West Grove, PA).

### Image and statistical analysis

Sample size and measurements are given as supplementary files for each accompanying figure. Analysis of the reporter constructs was performed on heterozygous reporter lines. Images were acquired with an Olympus FV1000 or a Zeiss Axio Imager VIS LSM880 confocal microscope with settings that avoided pixel intensity saturation. Representative images of Tv neurons being compared in figures were linear contrast enhanced together in Adobe Photoshop CS5 (Adobe Systems, San Jose, CA). All statistical analyses and graphing were performed using Prism 5 or Prism 8.0.1 (GraphPad Software, San Diego, CA). Normality of sample distribution was determined with Shapiro-Wilk normality tests. All multiple comparisons were done with One-Way ANOVA and a Tukey post-hoc test, or Student's two-tailed t-test when only two groups were compared. Mann-Whitney U-test was used when the samples were not normally distributed. Differences between groups were considered statistically significant when $p < 0.05$. Data are presented as either Mean ± Standard Deviation (SD).

## Quantification of reporter expression

Quantification of native EGFP reporter expression (without anti-GFP immunoreactivity enhancement) in late L3 larval VNCs was performed, in the context of anti-pMad immunoreactivity (to mark nuclei with active BMP-signaling). In all cases, five or more VNC were dissected and imaged for each genotype. All compared tissues were processed with the same reagents, imaged, and analyzed in identical ways. To quantitate reporter activity, we used Bitplane:Imaris v9.2 software (in Spots Mode) to identify reporter-positive nuclei in the VNC (excluding the brain lobes). When comparing control and pMad-binding site mutant genomic fragment reporters, we additionally assessed EGFP positive nuclei that were co-marked by pMad immunoreactivity (by mean intensity thresholding). Imaris settings were established independently for each set of reporters, in order to provide optimal 'spot' marking of a verifiable reporter and pMad co-immunoreactive nuclei, with minimal background fluorescence spot marking. Each image was further subtracted, manually, for spots that erroneously labeled background fluorescence.

## Gel shift assay

FLAG::Mad, Myc::Medea, and/or Tkv$^{QD}$ cDNA sequences were derived from previously described vectors (*Gao et al., 2005*) and subcloned into a pAc5.1/V5-His vector backbone (Thermo Fisher, Waltham, MA). *Drosophila* S2 cells were transfected at a density of $1.5 \times 10^6$ S2 cells per mL in a six well dish in 2 mL of media. A total of 2.4 ug of plasmid was used per well with each plasmid constituting 0.8 ug of the total. Total plasmid mass was kept constant by co-transfecting empty pAc5.1 when only one or two protein coding plasmids were used for an experimental condition. Transfections were performed with the XtremeGENE HD transfection kit according to the manufacturer's recommended protocol (Roche, Ontario, Canada). 48 hr after transfection, cells were harvested and lysed for gel shift assay. Cells were harvested in 15 mL tubes then pelleted by centrifugation (700 g at room temperature for 3 min). The pellets were re-suspended in PBS, transferred to 1.5 mL tubes, and centrifugation repeated as before. Cells were then resuspended in 90 µL ice-cold lysis buffer and the lysis reaction incubated on ice for 15 min. S2 cell lysis buffer contained 100 mM Tris HCl pH 7.6, 0.5% Tween-20, 1 mM DTT, and 1× Roche cOmplete ULTRA EDTA-free Protease inhibitor cocktail. The lysate was cleared by centrifugation at $16.2 \times 10^3$ g for 15 min at 4°C. The supernatant was aliquoted into new pre-chilled tubes in 30 uL volumes, snap-frozen in liquid nitrogen, and stored at −80°C until use. Oligonucleotides were synthesized and labeled with IRDye 700 by Integrated DNA Technologies (IDT, Indiana). DNA and protein binding was performed by incubating 20 µg of lysate protein with 1 µL of 50 nM IRDye 700-labeled probe in 20 µL of reaction buffer containing 25 mM Tris pH7.5, 35 mM KCl, 80 mM NaCl, 3.5 mM DTT, 5 mM MgCl$_2$, 0.25% Tween 20, 1 µg poly dIdC, 10% glycerol, and 1× Roche cOmplete ULTRA EDTA-free Protease inhibitor cocktail for 30 min at room temperature. If super shifts were performed, then 1 µg of mouse anti-MYC (clone 9E10, Sigma), mouse anti-FLAG (clone M2, Sigma), or mouse IgG (Sigma) was added and incubated for an additional 30 min at room temperature. The DNA-protein complexes were resolved on a 4% nondenaturing polyacrylamide gel for 1.5 hr at 70 V in 1× TGE buffer. After electrophoresis, the gel was imaged immediately by a Licor Odyssey Imager system (Lincoln, NE).

## Computational detection of BMP-LA instances

HOMER v4.10 software suite (*Heinz et al., 2010*) was used to scan the reference dm6 *Drosophila* genome for the BMP-LA (*GGCGCC(N5)GTAT*) motif. Base-specific PhastCons scores (*Siepel et al., 2005*) against 27-insect species were obtained from the UCSC genome browser (https://genome.ucsc.edu/) to annotate each motif instance with evolutionary conservation scores (*Supplementary file 1b*).

## Acknowledgements

The authors gratefully acknowledge Drs. Stefan Thor (Linkoping University, Sweden), Markus Affolter (University of Basel, Switzerland), Giorgos Pyrowolakis (University of Freiburg, Germany), Robin Vuilleumier, Hugh Brock, T Michael Underhill, Jacob Hodgson, Timothy O'Connor (University of British Columbia, Canada), and the Allan laboratory for their valuable assistance and comments. The authors gratefully acknowledge Drs. Konrad Basler and Johannes Bischof (University of Zurich,

Switzerland), Jonathan Benito-Sipos (Universidad Autónoma de Madrid, Spain), Allen Laughon (University of Wisconsin - Madison, USA), Stefan Thor (Linkoping University, Sweden), Esther Verheyen (Simon Fraser University. Canada), and the Bloomington *Drosophila* Stock Centre (Indiana, USA) for genetic, molecular, or immunological reagents.

## Additional information

### Funding

| Funder | Grant reference number | Author |
|---|---|---|
| Canadian Institutes of Health Research | MOP-98011 | Douglas W Allan |
| Canadian Institutes of Health Research | MOP-130517 | Douglas W Allan |

The funders had no role in study design, data collection and interpretation, or the decision to submit the work for publication.

### Author contributions

Anthony JE Berndt, Conceptualization, Data curation, Formal analysis, Validation, Investigation, Visualization, Methodology, Writing - original draft, Writing - review and editing; Katerina M Othonos, Conceptualization, Formal analysis, Validation, Investigation, Visualization, Methodology, Writing - original draft, Writing - review and editing; Tianshun Lian, Formal analysis, Validation, Investigation, Visualization, Methodology; Stephane Flibotte, Software, Writing - review and editing; Mo Miao, Raymond Y Cho, Justin S Fong, Seo Am Hur, Investigation, Writing - review and editing; Shamsuddin A Bhuiyan, Bioinformatics, Writing - review and editing; Paul Pavlidis, Software, Supervision, Writing - review and editing; Douglas W Allan, Conceptualization, Resources, Data curation, Formal analysis, Supervision, Funding acquisition, Validation, Investigation, Visualization, Methodology, Writing - original draft, Project administration, Writing - review and editing

### Author ORCIDs

Anthony JE Berndt (iD) https://orcid.org/0000-0002-0132-7393
Seo Am Hur (iD) http://orcid.org/0000-0003-4163-7182
Paul Pavlidis (iD) http://orcid.org/0000-0002-0426-5028
Douglas W Allan (iD) https://orcid.org/0000-0002-3488-8365

### Decision letter and Author response

Decision letter https://doi.org/10.7554/eLife.59650.sa1
Author response https://doi.org/10.7554/eLife.59650.sa2

## Additional files

### Supplementary files

• Supplementary file 1. Summary of wildtype and mutant Tv4-enhancer sequences tested for in vivo reporter activity. (A) The full length Tv4-enhancer was isolated from Oregon R and the sequence is shown here. (B-D) We introduced substitution mutants into the full length Tv4-enhancer to generate three mutant enhancers that were tested in *Figures 1* and *7*. (E,F) The Homeodomain Response Element (HD-RE) and BMP-Response cis-Element (BMP-RCE) were identified previously within the Tv4-enhancer. The sequences shown in E, F are a $6\times$ concatemer of the HD-RE (E) and $4\times$ concatemer of the BMP-RCE (F) enhancers used to drive EYFP in the $Tv^{HD-RE}$-EYFP and $Tv^{BMP-RE}$-EYFP transgenes, respectively in *Figure 2*. Bolded black letters are the Mad and Medea binding sites of the BMP-LA. Bolded red letters are the substitution mutations introduced. Green letters are the Restriction enzyme sites or Restriction scar sites in the concatemeric sequences of E,F. (b) List of 128 BMP-LA motifs in the *Drosophila* genome with an average PhastCons score over 0.55. The BMP-LA motifs are ranked according to their average PhastCONS score. The location of each motif is indicated

according to the August 2014 (BDGP Release 6 + ISO1 MT/dm6) Assembly. The first column indicates the assigned name for the BMP-LA as used throughout the manuscript or its location based on the nearest gene. Out of these 128 motifs, we found that 68% (87/128) is located within an intron or untranslated region (UTR), while the remaining 32% (41/128) are intergenic relative to the nearest gene. Blue BMP-LAs indicate those tested but not found to be BMP-dependent. Bolded red letters indicate the BMP-LA motifs in genomic fragments found to be BMP-dependent in third instar larva VNC. *Average evolutionary conservation score of a motif calculated based on the base-by-base conservation of each position in the motif. (c) Summary of BMP-LA containing genomic fragments tested for in vivo reporter activity. We prioritized 24 enhancers to test in vivo based on their proximity to genes expressed (4th column) in the third instar larvae VNC. Reporters were sorted based on the distance of the BMP-LA enhancer to the nearest CNS-expressed gene transcription start site (TSS). Bolded letters indicate the BMP-LA containing genomic fragments found to be *wit*-dependent in the VNC of third instar larvae. *Average evolutionary conservation score of a motif calculated on the base-by-base conservation of each position in the motif.

- Transparent reporting form

## Data availability

All data generated or analysed during this study are included in the manuscript and supporting files.

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
