## [Decision Letter]

[Editors’ note: the authors submitted for reconsideration following the decision after peer review. What follows is the decision letter after the first round of review.]

Thank you for submitting your work entitled "A suboptimized *cis*-regulatory BMP response element confers subtype-specific gene activation in *Drosophila* neurons" for consideration by *eLife*. Your article has been reviewed by three peer reviewers, and the evaluation has been overseen by a Reviewing Editor and a Senior Editor. The following individual involved in review of your submission has agreed to reveal their identity: Michael Eisen (Reviewer #1).

Our decision has been reached after consultation between the reviewers. Based on these discussions and the individual reviews below, we regret to inform you that your work will not be considered further for publication in *eLife* in its present state.

However as you'll see from the reviewers comments there was considerable enthusiasm for your work, but all three were of the opinion that the paper requires several additional experiments before we could consider it further at *eLife*. We appreciate the experiments suggested by the reviewers will take longer than the allowed time frame for revision, but in the case you would be willing to undertake them we would be happy to consider a new version of the manuscript at a future date.

Reviewer #1:

This is a simple, but well executed paper exploring the specificity of a 39bp BMP responsive element that confers neuronal specificity to the FMRFamide gene. I have no real issues with the experiments, as they convincingly establish that:

a) Medea is required to activate this construct

b) That the 39bp element is not dependent on Brinker or Schnurri

c) That activated Smad binds to the element in vitro in S2 lysates

d) That Smad binding is dependent on the non-canonical sequence GGCGCCNNNNNGTNT which is a poorer recruiter of Smad than the canonical response elements

e) That turning the suboptimal sequence into an optimal one eliminates proper activation and leads to broader ectopic activation.

Although there is now an established body of literature pointing to the importance of weak/suboptimal binding sites in tuning transcriptional responses, especially to graded signals, I think this paper makes a valuable contribution, and I support its publication more of less as is.

My only real tuck is in the interpretation. As the authors note, the loss of expression in the Tv4 neurons when the weak site is transformed into an optimal one, is, while not unprecedented, at least a bit surprising as it either means that stronger Smad binding is repressive for this element or it suggests that there is a dependence on some other factor. The latter hypothesis gains some weight from the observation that the A in the GTAT part of the Smad binding element is not required for Smad binding but is perfectly conserved across the genus. This deserved more attention in the discussion and potentially modifies the way in which these experiments fit in to the broader discussion of the role of suboptimal sites in tuning responses.

Reviewer #2:

This paper utilizes a very well-studied system of a single peptidergic neuron per hemisegment in the *Drosophila* embryo. Previous work from this and other labs have shown that the appropriate and timely specification of this FMRFa expressing Tv4 neuron requires co-ordinated activity of intrinsic genetic programmes (apterous, eya) with extrinsic, target derived cues (BMP pathway). The authors had previously identified an enhancer region of the FMRFa gene (Tv4 enhancer) that integrates these two genetic programmes and in this study they describe how BMP signaling acts at this enhancer.

BMP dependent gene activation can either be achieved directly through pMAD+Madea/Smad complex acting as transcriptional activators, or through de-repression via Brinker and Schnurri. Using mutants and genetic interactions combined with FMRFa expression and Tv4 reporter activity, (which recapitulates FMRFa expression in the Tv4 neuron), the authors show that the BMP pathway acts via the Madea/Smad mediated transcriptional activation pathway and does not involve Brinker and Schnurri. They then focus on the BMP-RE region of the Tv4 enhancer and identify a putative non-canonical binding site. Using S2 cells transfected with activated BMP receptor and tagged MAD and Madea/Smad along with gel shift assays the authors show that this non-canonical site is indeed essential for Madea/Smad recruitment and describe very thoroughly the sequences that are essential for Madea/Smad recruitment. They also demonstrate that the non-canonical nature of the site renders it sub-optimal in the recruitment of Madea/Smad. Finally, they demonstrate that this sub-optimal recruitment of the complex is essential for the expression of the Tv4 enhancer in the Tv4 neuron.

The experiments described are thorough, the writing is generally quite clear and the findings are interesting in the light of a diversity of outcomes of BMP signaling by changing binding site affinities and related compensatory nucleotide changes.

Specific Comments:

1) Figure 3 and related text: Although the figure and data supporting this section is quite clear, a little more attention to the writing would greatly enhance the interpretation of the data. For example, the authors might consider laying out individual genes and proteins, and their known interactions at both the transcriptional and protein interaction level at the top of the paragraph. A schematic illustrating this, along with possible outcomes of the genetic interactions would also benefit this section.

2) Figure 8: This figure in particular, would be more satisfying if an enlarged merged image was shown with the relevant counterstains and magnified insets of the cells depicting the individual channels were shown alongside.

Reviewer #3:

In their paper "A sub-optimized cis-regulatory BMP response element confers subtype-specific gene activation in Dropshila neurons," Berndt and colleagues provide a focused study on how a common BMP signal generates subtype-specific diversity. They found a minimal Smad-binding motif that matches a consensus motif except for a conserved nucleotide difference that attenuates Smad binding. Interestingly, they demonstrated that conversion to an "optimal" sequence resulted in ectopic reporter activity in other neurons. Together, this provides evidence that "sub-optimization" serves as a mechanism to provide specificity, giving additional evidence to the paradigm that low-affinity sequences are essential for precise gene expression.

While the paper is well done, combining both genetics and biochemistry, it would be great if the results could be expanded to generalize their findings. For example, with such a large consensus motif could the authors find additional motifs across the genome? Are they conserved? Are any of these motifs functional? This could be tested, for example, in existing Rubin Gal-4 lines. So, it could be done in a reasonable timeframe. Additionally, as noted below, the findings could be streamlined for easy reading. As it stands the manuscript is limited in scope and is very specialized.

Finally, "suboptimal." I recommend that the authors re-consider their use of "sub-optimized". As the authors demonstrate, the sequences are highly optimized for Tv4 neuronal expression. Furthermore, these "sub-optimal" sequences are conserved for millions of years, implying that they are optimal for expression. An alternative would be to call them low-affinity sequences, which is internally consistent across their results.

Figures:

1F, add shading as in panel 1B. I would also consider moving D-F to supplementary materials, to tighten up the flow of the paper.

Figure 2. Following from above, if panels D-F are moved to supplemental information, Figures 1 and 2 could be combined to demonstrate a conserved.

Figures 4-6. Much of this could be combined and even added to all of the above. See Crocker et al., 2015 for example; where Figure 2 has conservation, gel-shifts, and embryos with quantification.

References:

Gao et al., 2005 is repeated

For Crocker et al., 2016 cite the main cell paper (Crocker et al., 2015, Cell).

[Editors’ note: further revisions were suggested prior to acceptance, as described below.]

Thank you for submitting your article "A low affinity *cis*-regulatory BMP response element confers subtype-specific gene activation in *Drosophila* neurons" for consideration by *eLife*. Your article has been reviewed by two peer reviewers, and the evaluation has been overseen by a Reviewing Editor and K VijayRaghavan as the Senior Editor. The reviewers have opted to remain anonymous.

As you will see one of the reviewers is now more satisfied with the paper, but still asks from some slight editorial revisions. The other reviewer, a new reviewer, is taking a moderately supportive view of the manuscript. There is surely some editorial revision that can be easily done and increase satisfaction of the reviewer. The proposed experiment(s) are not essential, but would surely strengthen the paper.

Reviewer #1:

In its revised form, this manuscript addresses the suggestions and concerns that were raised in during its first submission.

Overall, it explores the details of specification of an FMRFa expressing “Tv4” neuron. For the appropriate expression of FMRFa in the Tv4 neuron, intrinsic genetic programmes (apterous, eya) are integrated with extrinsic, target derived cues (BMP pathway) at an enhancer called “Tv4” that the authors had previously identified. Here the authors show that the BMP pathway acts via the Madea/Smad mediated transcriptional activation pathway and does not involve Brinker and Schnurri mediated depression. They identify a non-canonical, low affinity BMP-response element, which is essential for the expression of the Tv4 enhancer in the Tv4 neuron. They next identify other such low affinity binding sites in the Drosphila genome and demonstrate its activity and dependence on Madea function.

This remains a manuscript of niche interest, however, as I had noted in the previous submission, the experiments are thorough and clearly presented. The writing is generally quite clear.

Reviewer #2:

Berndt et al. make an ambitious and thorough analysis of a degenerate BMP responsive element that direct FMRF expression. They demonstrate pMAD/medea to bind the motive bind with low affinity and activate expression from the element. Replacing the motive with a consensus element still direct expression to neurons but of not to the original FMRF neurons. A bioinformatics study identifies 103 conserved similar elements across the *Drosophila* genome. Production of reporters with selected motives directed expression to both pMAD positive and negative neurons and glia. Even if the take is ambitious and low affinity motives in my view often marginalised, I have some concerns about the general applicability and advances made here.

Major concerns:

The authors state that this study will be important for how bioinformatics view low affinity motives. Even if the analysis is ambitious, the outcome is not encouraging for any predictive use. The cis regulatory regions containing the degenerate motive does not restrict expression to a predictable set of neurons or to neurons in general. The question that remains is if regions with perfect motives would do better or worse in restricting expression to these neurons.

For other signalling systems similar low affinity motives provide instructive values in target gene regulation a phenomenon described as activation insufficiency and important for a signalling system not to dominate the regulation of the particular gene (for review see Barolo and Posakony, 2002). This raises the question if evolution placed this motive strength there to fit a certain signalling level. A question that could simply be addressed by increasing (dominant active, ligand overexpression) or decreasing (heterozygote ligand mutants, ts mutants) signaling activity and observe the reporter. In the current format the signalling twist is there but only for the FMRFamide gene and without any predictive input.

It is argued that “this manuscript defines how common intercellular signals so exquisitely direct subtype-specific gene expression profiles”. There are low affinity instructive models also for neurons. For example, in *Drosophila* the olfactory system the interaction between several low affinity motifs and chromatin has been modelled and tested to be required to restrict odorant receptor expression to a single neuron class. (Jafari and Alenius, 2015, González and Jafari et al., 2018). However, these factors are not related to known signaling systems one can argue and like that these findings become an advance. Nevertheless, the authors need to make the discussion more comprehensive to make that claim and define the mechanism in more detail.

Thus, the study is well performed and a nice addition to the list of similar studies but with little signalling context, predictive power or computer model testing the study is not a major advance compared to what already is published.

---

## [Author Response]

[Editors’ note: the authors resubmitted a revised version of the paper for consideration. What follows is the authors’ response to the first round of review.]

The only additional experiments that were requested are captured in the comments of reviewer 3:

“While the paper is well done, combining both genetics and biochemistry, it would be great if the results could be expanded to generalize their findings. For example, with such a large consensus motif could the authors find additional motifs across the genome? Are they conserved? Are any of these motifs functional? This could be tested, for example, in existing Rubin Gal-4 lines”

We have now performed the requested experiments and include the results in this re-submitted manuscript. The data confirm the hypothesis that the low-affinity BMP-RE motif we report in this manuscript is a widely deployed, functional BMP-response element.

1) We now provide a list of all motifs matching the low-affinity BMPresponse element (BMP-RE) motif in the *D. melanogaster* genome and provide their PhastCons scores for motif conservation across all other sequenced *Drosophila* species (Supplementary File 1B) These analyses were performed using the HOMER suite of tools for Motif Discovery.

2) Based on these data, we selected 24 genomic regions forfunctional characterization of candidate motif activity

(Supplementary file 1C). We cloned ~2kb genomic fragments, inclusive of a single candidate BMP-RE motif, upstream of a nuclearlocalized GFP reporter and integrated these into the attP2 site of *Drosophila* by phiC31-integrase dependent transgenesis. These were examined for expression in BMP-activated neurons.

3) We observed that 10 of these were expressed in BMP-activated neurons and that the expression of the majority of those (7/10) was significantly reduced in wit nulls, wherein neuronal BMP signalling is abrogated.

4) Based on these results, we performed a second round of transgenesis to confirm that the BMP-RE motif itself is essential for expression, by re-testing reporter expression after mutation of the BMP-RE so as to prevent pMad/Medea binding. The significant reduction in expression for these (that was mostly comparable to the loss of expression of wildtype genomic fragments in wit nulls), confirms that the BMP-RE itself is functional.

Reviewer #1:[…]My only real tuck is in the interpretation. As the authors note, the loss of expression in the Tv4 neurons when the weak site is transformed into an optimal one, is, while not unprecedented, at least a bit surprising as it either means that stronger Smad binding is repressive for this element or it suggests that there is a dependence on some other factor. The latter hypothesis gains some weight from the observation that the A in the GTAT part of the Smad binding element is not required for Smad binding but is perfectly conserved across the genus. This deserved more attention in the discussion and potentially modifies the way in which these experiments fit in to the broader discussion of the role of suboptimal sites in tuning responses.

We thank the reviewer for emphasizing the need to enhance this discussion. We have now expanded this discussion.

Reviewer #2:[…]Specific Comments:1) Figure 3 and related text: Although the figure and data supporting this section is quite clear, a little more attention to the writing would greatly enhance the interpretation of the data. For example, the authors might consider laying out individual genes and proteins, and their known interactions at both the transcriptional and protein interaction level at the top of the paragraph. A schematic illustrating this, along with possible outcomes of the genetic interactions would also benefit this section.

This has now been performed (schematic in Figure 2H-I)

2) Figure 8: This figure in particular, would be more satisfying if an enlarged merged image was shown with the relevant counterstains and magnified insets of the cells depicting the individual channels were shown alongside.

This has now been performed (Figure 5D-E)

Reviewer #3:[…]Finally, "suboptimal." I recommend that the authors re-consider their use of "sub-optimized". As the authors demonstrate, the sequences are highly optimized for Tv4 neuronal expression. Furthermore, these "sub-optimal" sequences are conserved for millions of years, implying that they are optimal for expression. An alternative would be to call them low-affinity sequences, which is internally consistent across their results.

We thank the reviewer for this suggestion. This phrase is indeed more appropriate.

Figures:1F, add shading as in panel 1B. I would also consider moving D-F to supplementary materials, to tighten up the flow of the paper.Figure 2. Following from above, if panels D-F are moved to supplemental information, Figures 1 and 2 could be combined to demonstrate a conserved.Figures 4-6. Much of this could be combined and even added to all of the above. See Crocker et al., 2015 for example; where Figure 2 has conservation, gel-shifts, and embryos with quantification.References:Gao et al., 2005 is repeatedFor Crocker et al., 2016 cite the main cell paper (Crocker et al., 2015, Cell).

These have all been corrected.

[Editors’ note: what follows is the authors’ response to the second round of review.]

Reviewer #2:Major concerns:The authors state that this study will be important for how bioinformatics view low affinity motives. Even if the analysis is ambitious, the outcome is not encouraging for any predictive use. The cis regulatory regions containing the degenerate motive does not restrict expression to a predictable set of neurons or to neurons in general.

These comments raise important issues that we had not addressed adequately in our previous manuscript. We have softened any statements about what our work says regarding computational approaches to low affinity motif discovery, focusing solely on the fact that the BMP-LA motif can be used in such an approach.

We would argue that our studies do support the predictive value of the BMP-LA motif in the *in silico* discovery of BMP-responsive *cis*-elements, as supported by our identification of novel BMP-responsive genomic regions. However, the reviewer’s comments alerted us to an unintended interpretation of our conclusions, implying that the BMP-LA motif inherently confers restricted enhancer activity to predictable sets of neurons. We thank the reviewer for bringing this to our attention, because this was not intended. In response, we have restructured the Introduction and Discussion to more precisely state our view. This is most explicitly stated in a section starting with the following statement “We propose that the BMP-LA does not singularly specify Tv4 neuron activity, but rather imposes a requirement for pMad/Medea cooperative interactions with co-regulators or subtype-specific transcription factors bound to adjacent motifs for combinatorial activation of FMRFa expression.” This is followed by extensive discussion to explain our rationale for this conclusion and how the BMP-LA generates subtype-specific neuronal expression, based on evidence from this study and precedent from previous work. We also added a related edit in the Results section. We feel that this new discussion has improved our manuscript and clarifies our conclusions, while better addressing a model for how the BMP-LA motif functions in enhancers active across many cell types in neurons and across fly tissues.

The question that remains is if regions with perfect motives would do better or worse in restricting expression to these neurons.

We have incorporated a more thorough discussion of how we believe the BMP-LA motif generates cell-specific expression (as stated above). Our previous publication on the specification of gene expression within Tv4 neurons (1) identified two essential, non-redundantly acting, *cis*-elements that encode Tv4-neuron specification, an Apterous-responsive element (that we termed the HD-RE) and the BMPRCE element containing the BMP-LA identified herein. This is outlined at the start of the Results section. This HD-RE contains a “perfect” Apterous-binding motif, but Apterous is not sufficient alone for its expression, leading us to assume that other unidentified transcription factors are required to act at this *cis*-element. As these remain unidentified it is not possible to say whether their motif within the HD-RE is perfect or not. Regardless, while both *cis*elements encode Tv4-neuron specificity (which is only observed when we concatemerized each *cis*-element), the HD-RE is oblivious to BMP signaling; thus, the BMP-dependent activation of the *FMRFa* gene is entirely encoded within the BMP-RCE, via the BMP-LA. The combined activation of both motifs is required for *FMRFa* expression. Importantly, when we swap the BMP-LA for a high affinity BMP-AE motif, we do so in the context of the full-length Tv4 enhancer for the *FMRFa* gene (containing this HD-RE *cis*-element). Therefore, the more widespread ectopic expression of this reporter in motor neurons and its exclusion from the Tv4 neurons is driven by the more promiscuous BMP-AE motif, and no other *cis*-element with the Tv4 enhancer seems able to compensate or correct. Therefore, we feel confident in concluding that the BMP-LA plays essential roles in both conferring a BMP requirement for *FMRFa* expression, and also conferring Tv4 neuron specification to the full-length enhancer. We have added this discussion in paragraph four of the Discussion.

For other signalling systems similar low affinity motives provide instructive values in target gene regulation a phenomenon described as activation insufficiency and important for a signalling system not to dominate the regulation of the particular gene (for review see Barolo and Posakony, 2002). This raises the question if evolution placed this motive strength there to fit a certain signalling level. A question that could simply be addressed by increasing (dominant active, ligand overexpression) or decreasing (heterozygote ligand mutants, ts mutants) signaling activity and observe the reporter. In the current format the signalling twist is there but only for the FMRFamide gene and without any predictive input.

These are important points that we have expanded upon in our discussion. We would certainly agree that most natural BMP-RE instances exhibit activation insufficiency, regardless of whether they are low affinity or not. Our previous publication (1) offered a detailed study of the requirement for the numerous transcription factors that were combinatorically required for Tv4 enhancer of the *FMRFa* gene. However, in this manuscript we are attempting to show that the function of BMP responsive DNA motifs in an enhancer is not only to add a generic ability to respond to BMP signalling in any BMP-activated neuron subtype (regardless of its affinity), but that the low affinity of this BMP-RE was required to participate instructively in determining the subset of BMP-activated neurons an enhancer is active, in combination with other local transcription factors.

In previous publications, we tested the impact of heterozygosity for BMP pathway components and also overexpression of the pertinent signaling ligand Gbb, and also activated type I BMP-receptors, on expression of the *FMRFa* gene and also on reporters containing the BMP-LA (1–4). In all cases, no ectopic expression of the gene or the reporter was observed by BMP hyperactivation, unless we also co-misexpressed other transcription factors. Therefore, we conclude that responsiveness to a specific signal strength is not a major pressure that led to the specific sequence of the BMP-LA. These points have now been added to the Discussion section starting in paragraph three.

It is argued that “this manuscript defines how common intercellular signals so exquisitely direct subtype-specific gene expression profiles”. There are low affinity instructive models also for neurons. For example, in *Drosophila* the olfactory system the interaction between several low affinity motifs and chromatin has been modelled and tested to be required to restrict odorant receptor expression to a single neuron class. (Jafari and Alenius, 2015, González and Jafari et al., 2018). However, these factors are not related to known signaling systems one can argue and like that these findings become an advance. Nevertheless, the authors need to make the discussion more comprehensive to make that claim and define the mechanism in more detail.

We thank the reviewer for this valuable suggestion, and agree that adding this discussion strengthens the paper. The Introduction and Discussion have been substantially restructured and rewritten to provide a more comprehensive contemplation of these points. These highly relevant studies have been included in the Discussion.